# Mechanical Load and Piezo1 Channel Regulated Myosin II Activity in Mouse Lenses

**DOI:** 10.3390/ijms23094710

**Published:** 2022-04-24

**Authors:** Ariana Allen, Rupalatha Maddala, Camelia Eldawy, Ponugoti Vasantha Rao

**Affiliations:** 1Department of Ophthalmology, Duke University School of Medicine, Durham, NC 27710, USA; ariana.allen@duke.edu (A.A.); rupa.maddala@duke.edu (R.M.); camelia.eldawy@duke.edu (C.E.); 2Department of Pharmacology and Cancer Biology, Duke University School of Medicine, Durham, NC 27710, USA

**Keywords:** lens, piezo channel, stiffness, myosin II, calpain, mechanotransduction, cataract

## Abstract

The cytoarchitecture and tensile characteristics of ocular lenses play a crucial role in maintaining their transparency and deformability, respectively, which are properties required for the light focusing function of ocular lens. Calcium-dependent myosin-II-regulated contractile characteristics and mechanosensitive ion channel activities are presumed to influence lens shape change and clarity. Here, we investigated the effects of load-induced force and the activity of Piezo channels on mouse lens myosin II activity. Expression of the Piezo1 channel was evident in the mouse lens based on immunoblot and immufluorescence analyses and with the use of a Piezo1^-tdT^ transgenic mouse model. Under ex vivo conditions, change in lens shape induced by the load decreased myosin light chain (MLC) phosphorylation. While the activation of Piezo1 by Yoda1 for one hour led to an increase in the levels of phosphorylated MLC, Yoda1 treatment for an extended period led to opacification in association with increased calpain activity and degradation of membrane proteins in ex vivo mouse lenses. In contrast, inhibition of Piezo1 by GsMTx4 decreased MLC phosphorylation but did not affect the lens tensile properties. This exploratory study reveals a role for the mechanical load and Piezo1 channel activity in the regulation of myosin II activity in lens, which could be relevant to lens shape change during accommodation.

## 1. Introduction

The transparent ocular lens plays a crucial role in vision by focusing light onto the retina, and aberrations in lens transparency and mechanical properties can lead to visual impairment. Cataract formation and presbyopia represent the two most prevalent forms of visual impairment in humans that are caused by the loss of clarity and focusing abilities, respectively, in the transparent ocular lens [1,2]. Although therapeutic options are a desirable alternative to surgical treatments for these common visual defects, we do not currently possess a thorough understanding of the fundamental mechanisms regulating and maintaining the architecture, clarity, and mechanical properties of lens to inform the development of targeted therapies for cataracts and presbyopia.

For example, while the lens shape is changed reversibly during accommodation, the regulation of this process is poorly understood. Although ciliary muscle contraction and relaxation play a critical role in lens shape change during accommodation, by controlling the tension of the zonules of Zinn, which are physically connected to the lens, our understanding of the biochemical pathways involved in this process has remained elusive [3,4]. Moreover, when the lens undergoes deformation and recovers its normal shape during accommodation, the lens cells are expected to encounter and sense changes in the mechanical force and membrane stretch induced by altered zonule tension and ciliary muscle contraction/relaxation [4,5]. Cells utilize the process of mechanotransduction to sense and respond to mechanical stimuli by converting them to biochemical signals that elicit specific cellular responses [6]. Lens fibers have been shown to contain high levels of phosphorylated myosin light chains, the regulatory subunit of myosin II [7]. Myosin II is a motor protein that binds to actin to regulate actomyosin stress fiber contraction and relaxation, and generates traction force [8]. Myosin II is a well-characterized mechanosensitive protein involved in the regulation of various cellular processes [9]. Mutations in non-muscle myosin IIA are associated with lens defects and cataract formation, indicating a requirement for the structural and functional integrity of myosin II in the maintenance of lens architecture and function [10,11]. However, we know very little about the role of myosin II activity during accommodation-induced changes in lens shape [12].

Additionally, the bulk of the lens is comprised of elongated, ribbon-like differentiating and differentiated fiber cells, whose predominant components are the plasma membrane and membrane proteins [13,14]. Moreover, although it is well recognized that membranes bound channel proteins and transporters, and the membrane skeleton organization of lens fibers play a vital role in lens growth and function and in cataract development, we have minimal understanding of the role of channel proteins in lens shape change during accommodation [13,14,15]. The lens contains not only gap junctions and hemi channels made up of connexins and water channels but also L-type channels and transient receptor protein ion channels including the TRPV and TRPM channels [5,15,16,17]. Regulation of calcium influx by some of these channels is expected to influence actomyosin contraction and relaxation. Indeed, inhibition of L-type channel activity has been shown to impair myosin light chain (MLC) phosphorylation and thus myosin II activity in the lens [18]. Inhibition of calcium-dependent MLC kinase has also been shown to impair lens clarity and stiffness [7,12]. It is now well recognized that many tissues and cells express Piezo channels, which are nonselective cation channels activated by mechanical stimuli (including touch, membrane tension, traction force, and shear stress) that are involved in the regulation of various cellular activities and calcium influx and disease mechanisms [19,20,21,22]. However, the expression, distribution, and role of Piezo channels in lens functions are not well known [23].

To gain insights into the plausible involvement of mechanotransduction in lens shape change, and to address how lens shape change and Piezo channel activity might influence myosin II activity in lens, in this study we determined the effects of lens shape change and Piezo channel activity on the myosin II activity, stiffness, and clarity of mouse lenses.

## 2. Results

### 2.1. Lens Deformation Induced by Mechanical Loading Decreases Myosin II Activity

To determine the effects of external-mechanical-loading-induced lens shape changes on myosin II activity, a glass stopper (weighing ~3.5 g) was placed on a freshly enucleated lens derived from 3- to 4-week-old mice for a period of sixty seconds prior to lens fixation with trichloroacetic acid (TCA) as described in the Methods section. As shown in Figure 1, weight loading exerted by a glass stopper led to an obvious deformation of the lens with a significant increase (by ~24%, *n* = 6) in its width without rupture, compared to the control lenses. The deformation/shape change induced by mechanical loading did not affect lens transparency. These load-induced deformed lenses and their respective control lenses were analyzed for changes in their levels of phosphorylated MLC by immunoblot analysis. MLC is a regulatory subunit of myosin II activity, with MLC phosphorylation and dephosphorylation activating and inhibiting myosin II activity, respectively [24,25]. The deformed lenses revealed a significant decrease (>65%, *n* = 7) in the levels of phospho-MLC compared to control lenses which were not subjected to load-induced shape changes (Figure 1). Phospho-MLC levels were normalized to β-actin, and β-actin levels were found to be comparable between the deformed and control lenses.

In addition to experiments conducted using glass stoppers (Figure 1), glass coverslips were also utilized to induce lens deformation. For this, we placed either 5 (1.07 g total weight) or 10 (2.14 g of total weight) coverslips on two different sets of lenses (4 weeks old) for a period of 60 s. Following lens compression, one half of the lenses from each set were treated immediately with TCA following removal of the glass coverslips, while the other half were treated with TCA 60 s after removing the coverslips, prior to determining the levels of phospho-MLC and total MLC. Similar to the results from the glass stopper treatment described above, lens compression induced with both 5 and 10 glass coverslips for 60 s was associated with an obvious decrease (ranging from 40 to 47%) in the levels of phospho-MLC compared to uncompressed lenses (Figure 2, *n* = 5 experimental replicates). There is a slight but significant increase in phospho-MLC levels of ~25% upon removal of the glass coverslips (*n* = 5 coverslips) during the resilience period (Figure 2B). However, overall, in the compressed lenses upon removal of the glass coverslips (both 5 and 10), there was still a marked decrease in the levels of phospho-MLC (ranging from 20 to 60%) compared to the control lenses. Total MLC levels were not different between the compressed and uncompressed (control) lenses. Collectively, the results derived from the lens compression (using either a glass stopper or coverslips) experiments reveal that external-load-induced mechanical deformation of lens is associated with decreased intracellular myosin II activity.

### 2.2. Expression and Distribution of Piezo1 Channel in the Mouse Lens

After finding that external compression stress can influence intracellular myosin II activity in the lens, we sought to gain molecular insights into the possible mediators involved in the transduction of mechanical loading stress into changes in the activity of intracellular calcium-dependent myosin II. To this end, we explored a plausible role for Piezo channels, which are well-characterized mechanosensitive transmembrane channel proteins known to participate in the regulation of myosin II activity [26]. Since only little is known about the expression of Piezo channels in the lens [23], we initially examined the RNA-seq-based transcriptome profile of P30 mouse lens data that we recently generated [27]. In two independent samples (each sample was derived from the pooling of several lenses), we detected the presence of Piezo1 and Piezo2 transcripts. The relative abundance of Piezo1 (Relative Signal Intensity—RSI: 1680 ± 82.03) expression was found to be much higher (by ~20 fold) than that of Piezo2 (RSI: 80.5 ± 3.57) based on the average values of two samples. These RNA-seq-based results for the expression profile of Piezo channels in the lens tissue were independently confirmed by RT-PCR and qRT-PCR analyses using RNA derived from both P1 and P30 mouse lenses. Figure 3A shows the RT-PCR-based confirmation of Piezo1 and Piezo2 expression in P1 and P30 mouse lenses. Consistent with the RNA-seq-based findings, the relative expression level of Piezo1 was much higher than that of Piezo2 in both the P1 and P30 lenses. This was further confirmed by qRT-PCR analysis using samples derived from the P30 mouse lenses (Figure 3B).

The total lysates (800× *g* supernatants; 75 µg protein) derived from the P1, P14, and P16 mouse lenses were used to determine the presence of the Piezo1 protein by immunoblot analysis. Proteins separated on a gradient gel of 4–20% acrylamide showed immunopositive bands against Piezo1 polyclonal antibodies with an expected molecular mass of >250 kDa and a positive band at >150 kDa. There was also a prominent immunopositive band at >75 kDa in the P1 and P14 lens samples analyzed, with relatively reduced levels in the P16 lenses (Figure 3C). Interestingly, the Piezo1 immunopositive bands of >250 and >75 kDa were present predominantly in the lens fiber mass samples (P21 and P27) compared to the lens epithelium (P21) as shown in Figure 3D. Consistent with these results, immunofluorescence analysis of Piezo1 in the P1 mouse lens (the sagittal plane of the cryosection) revealed a distribution localized predominantly to lens fibers relative to the epithelium (Figure 3E,F). In contrast to immunofluorescence detected using Piezo1 antibodies (Figure 3E,F), lens sections stained with secondary antibodies alone did not show detectable fluorescence (Figure 3G), confirming the specificity of Piezo1 antibody staining. In contrast to Piezo1, the Piezo2 protein was undetectable by immunoblot analysis in the lens lysates derived from P1, P14, and P25 wild-type mice (Appendix A).

Since multiple Piezo 1 immunopositive bands were detected in wild-type mouse lens lysates using the described polyclonal Piezo1 antibody, as shown in Figure 3C,D, a Piezo1^-tdT^ mouse model (P30) expressing a fusion protein of Piezo1 and tdTomato was used to obtain additional and independent evidence for the expression and distribution of Piezo1 in the mouse lens. Similar to what was found in the wild-type lenses, the expression of the Piezo1^-tdT^ fusion protein (expected molecular mass of above 250 kDa) was detected in lens homogenates from P30 Piezo1^-tdT^ mice but not in the wild-type lens, with the positive control (lung tissue from Piezo1^-tdT^ mice) exhibiting a robust expression of the Piezo1^-tdT^ fusion protein (Figure 4A). Moreover, as in the case of wild-type lenses, the Piezo1 fusion protein was also detected predominantly in the fiber cell lysates compared to the lens epithelial lysates (Figure 4B) from Piezo1^-tdT^ mice. Immunofluorescence analysis also revealed the distribution of the Piezo1^-tdT^ fusion protein to the lens fibers with localization to both the short and long arms of the hexagonal fibers (Figure 4C, left and middle panels represent low and high magnifications, respectively). The lens section derived from the Piezo1P1^-tdT^ mouse stained with second antibodies alone (Alexa Fluor 488) showed very weak or no fluorescence (Figure 4C; right panel). The images shown in Figure 4C were from the outer cortical region of Piezo1^-tdT^ mouse lenses. Appendix A shows no specific immunofluorescence for the Piezo1^-tdT^ fusion protein in the lens epithelium of Piezo1^-tdT^ mice.

### 2.3. Piezo1-Activation-Induced Effects on Myosin II Activity and Clarity of the Mouse Lens

To determine the effects of Piezo1 activation on lens transparency and myosin II activity, lenses derived from P30 mice and maintained in culture were treated with 10 and 25 µM Yoda1, an agonist of Piezo1, and monitored for changes in transparency for 24 h. Lenses treated with 25 µM Yoda1 exhibited haziness starting at the 1 h interval, with the haziness increasing progressively with time, demonstrating slight swelling and a slight nuclear opacity by 24 h, compared to the control lenses as shown in Figure 5A. In contrast, lenses treated with 10 µM Yoda1 exhibited haziness and slight opacification only after 24 h of treatment compared to the control lenses (not shown), indicating that Yoda1 exerts dose- and time-dependent effects on lens clarity. The wet weights of lenses treated with 25 µM Yoda1 were significantly increased after 6 h (increase of 12.8%, control = 4.36 ± 0.197, test = 4.92 ± 0.197 mg/lens, *n* = 4) and after 24 h (increase of 42.66%, control = 4.3 ± 0.179, test = 5.9 ± 0.173 mg/lens; *n* = 7).

After recording that sustained Piezo1 activation affects lens clarity in association with lens swelling, we evaluated the Yoda1-treated lenses for changes in their myosin II activity by determining the levels of phosphorylated MLC and total MLC by immunoblotting analysis. Interestingly, the levels of phospho-MLC were significantly elevated (~60%; *n* = 6) at the 1 h interval in lenses treated with Yoda1 (Figure 5B). In lenses treated with Yoda1 for 6 and 24 h, however, there was a significant and progressive decrease (by >50 to 80%, respectively, *n* = 6) in the levels of phospho-MLC compared to the control lenses (Figure 5B). The levels of total MLC, however, were comparable between the Yoda1-treated versus the control lenses throughout the course of drug treatment, as shown in Figure 5B.

### 2.4. Yoda1 Treatment Induces Calpain Activity and Membrane Protein Proteolysis in Ex-Vivo Lens

To gain further insight into the Yoda1-induced effects on lens clarity and myosin II activity, we determined the activity of lens calpain, which is calcium dependent, since Piezo1 activation by Yoda1 is expected to increase calcium influx [26,28,29]. Lenses treated with 10 µM Yoda1 for 24 h exhibited a significant increase (of ~69%, *n* = 5) in calpain activity compared to the control lenses (Figure 6A). Based on these findings, we also examined the integrity of the lens membrane protein fraction by SDS-PAGE analysis of the membrane-enriched protein fraction derived from the Yoda1-treated (10 or 25 µM for 24 h) and control lenses. As shown in Figure 6B, the Coomassie-blue-stained SDS-PAGE profile of the membrane-enriched fraction from Yoda1-treated lenses showed either a complete loss or severely reduced staining of select proteins relative to the profile exhibited by the control lenses (Figure 6B, indicated with arrows). Collectively, these results indicate an activation of calcium-dependent calpain activity leading to plasma membrane protein degradation in ex vivo lenses in response to prolonged activation of Piezo1.

### 2.5. Piezo1-Inhibition-Associated Changes in the Lens Myosin II Activity

To determine the effects of Piezo1 inhibition on lens transparency, lenses derived from P30 mice were treated for 48 h with GsMTx4 (2.44 µM), a specific inhibitor of Piezo1. During the course of this drug treatment, the lenses remained largely transparent, with no significant change in wet weight (4.33 ± 0.156, mg/lens, *n* = 5) compared to the control (4.14 ± 0.036, *n* = 5) lenses (Figure 7A). Evaluation of phospho-MLC and total MLC levels revealed that GsMTx4-treated (48 h) lenses showed a significant decrease (by ~38%) in their levels of phospho-MLC compared to the control lenses (Figure 7B,C). There was no significant difference in the levels of total MLC in GsMTx4-treated lenses compared to the control lenses (Figure 7B,D).

### 2.6. Piezo1 Channel Inhibition Does Not Disrupt Normal Tensile Properties of the Lens

To test whether Piezo1 channel activity effects the tensile properties of the lens, we performed a lens stress/strain analysis of the GsMTx4-treated (piezo1 inhibitor, 2.44 µM for 48 h) and control lenses using a microstrain analyzer. As shown in Figure 8, changes in Young’s modulus calculated from the lens prerupture stress/strain slopes were not found to be different between the GsMTx4-treated and control lenses, indicating that Piezo1 inhibition under the described conditions does not impact the tensile characteristics of the lens.

## 3. Discussion

Although accommodation of the ocular lens is associated with deformability resulting from the altered mechanical tension of the zonules of Zinn induced by ciliary muscle contraction and relaxation [3,4,30], we know very little about how the lens transduces mechanical force into biochemical responses during this process. Towards this end, our results reveal that the expansion of the mouse lens width induced by force/stress resulting from mechanical loading decreases calcium-dependent myosin II activity in ex vivo experiments. Moreover, this study not only presents evidence for the expression of the evolutionarily conserved mechanical-force-sensitive Piezo channels in lens but also a role for Piezo channels in lens myosin II activity. Since myosin II plays a crucial role in regulating actomyosin assembly, contractile characteristics, traction force, cell adhesive interactions, and cell morphology, it is plausible that Piezo channels influence lens shape change in accommodating species by virtue of regulating calcium influx, myosin II activity, and actomyosin assembly and organization under a mechanical force.

The lens expresses both non-muscle myosin IIA and IIB and calcium-dependent MLC kinase, whose activities are required for the lens architecture and function [7,10,12]. Importantly, the inhibition of myosin II activity has been shown to impact chicken lens stiffness, indicating the involvement of myosin II and actomyosin assembly and organization in the regulation of lens tensile characteristics [12]. However, whether external-force-induced compression of the lens influences myosin II activity has not been addressed or understood. Although this study used mouse lenses, which are relatively hard and do not accommodate compared to the human lens, mechanical-loading-induced compression of the mouse lenses led to a significant change in the lens shape, especially a marked expansion in the width without rupture. Our study focused not only on compression-induced changes in MLC phosphorylation, but also evaluated the MLC phosphorylation status during a resilience phase (removal of the lens-deformation-inducing mechanical load). Interestingly, the force-induced decrease in myosin II activity appears to be reversible to some extent upon removal of the load in lenses subjected to a smaller load (using five glass coverslips), but not in lenses compressed using 10 glass coverslips. Although this study did not assess or quantify the reversible changes in lens width with and without glass coverslips placed on them, in future studies, we plan to quantitate and correlate changes in lens shape with myosin II activity during compression and resilience. However, these initial findings suggest that milder reversible changes in lens shape might be associated with reversible changes in the levels of phospho-MLC. Whether this response would be much more evident and prominent in accommodative lenses relative to the mouse lens needs to be evaluated in future studies using human or soft lenses from other species. Moreover, under ex vivo conditions, mechanical loading has been shown to irreversibly affect mouse lens fiber tip attachment to the capsule and epithelium, indicating that during the resilience phase, not all characteristics revert back to the preloading conditions [31]. However, the observation that lens compression induced by external mechanical loading influences intracellular myosin II activity gives clear evidence for the activation of biochemical pathways related to mechanotransduction in the intact lens. Therefore, in future studies, it is important and necessary to investigate whether accommodation-associated changes in the lens shape, induced by ciliary muscle contraction and zonule tension, are linked to the change in myosin II activity through mechanotransduction. These possibilities raise the next question of how mechanical force is transduced to the change in the myosin II activity in the lens.

To gain insight into the mechanisms by which external force regulates calcium-dependent intracellular myosin II activity in the lens, we focused on the Piezo channels. These channels are recently discovered, well-characterized, mechanosensitive nonselective cation channel proteins involved in mediating responses to touch, shear, and mechanical stresses, with well-recognized roles in various physiological and pathological processes [19,21]. Piezo channels are large proteins containing multiple transmembrane domains, with no known homology to any other proteins [19]. Activation of these channels induces calcium influx and regulates various calcium-dependent cellular activities [21,29]. Importantly, the membrane bilayer tension, actin cytoskeletal organization, cell adhesion, and myosin-induced traction force have also been found to influence piezo channel gating in different cell types [8,32,33]. Based on these characteristics of the Piezo channels, we examined expression of these proteins in the mouse lens. Expression of Piezo1 was found to be much higher compared to that of Piezo2, with a prominent distribution of the former protein to lens fibers relative to the lens epithelium. Interestingly, Piezo1 appears to undergo post-translational changes in the lens, including proteolysis, because in immunoblot analysis, we detected a prominent protein band at 75 kDa in addition to an expected protein band above 250 kDa corresponding to the native Piezo1 protein. This 75-kDa species, which was detected by the Piezo1 antibody used in our study, was prominent in the fiber cell lysates but absent in the lens epithelial lysates. Post-translational changes in Piezo1 appear to not be specific to the lens alone because many commercially available Piezo1 antibodies have been found to detect a prominent band around ~70–75 kDa based on information included in technical datasheets from the companies Thermofisher, ProteinTech, and Novusbio.

Interestingly, Yoda1-stimulated activation of the Piezo1 channel led to an increase in lens myosin II activity. Although we did not determine the change in intracellular calcium levels in this study, stimulation of calcium influx by Piezo channel activation has been reported in several cell types [21,26,28,29]. Therefore, we believe that Piezo-channel-dependent calcium influx stimulates MLC activity and thereby leads to an increase in myosin II activity in the lens. This was partly evident based on the observation of the activation of calpain activity in the Yoda1-treated lenses. Increased myosin II activity was evident only at the 1 h interval in lenses treated with Yoda1, but was decreased in lenses treated for 6 and 24 h. A noteworthy observation is that there was a significant and progressive increase in the wet weight of lenses treated with Yoda1 for 6 and 24 h. This suggests that the initial agonist-mediated activation of Piezo channels induces calcium influx, which appears to activate calpain leading to increased proteolysis of the membrane proteins in the lens. These changes very likely account in whole or partially for the haziness and opacification of the observed lenses. Therefore, in the 6 and 24 h Yoda1-treated lenses, it is possible that despite the activation of Piezo channel activity and calcium influx, an increase in haziness, opacification, and membrane protein degradation collectively and secondarily impairs myosin II activity. However, it is important to interpret these results with caution, because the effects of Piezo channel activation by Yoda1 in lenses maintained under ex vivo conditions might not be identical to those in in vivo lenses, since the regulation of Piezo and other channel protein activities is expected to be much more dynamic under physiological conditions relative to the effects determined under long hours of sustained activation in ex vivo lenses. Therefore, the opacification noted in ex vivo lenses under the sustained activation of Piezo channel activity could be specific to the experimental conditions described. An additional aspect for consideration is that other mechanosensitive channels including the TRPV4 channel may also be involved in mediating the effects of the force-induced lens shape during accommodation [5,17]. Moreover, the subcellular distribution of the TRPV4 channel has been shown to be influenced by zonule tension in mouse lenses under ex vivo culture conditions [5]. Whether zonule tension also affects the Piezo1 channel’s cellular distribution and activity and myosin II activity in the lens needs to be investigated in future studies.

Finally, our observation on the effects of Piezo channel inhibition on myosin II activity in contrast to the piezo-activation-induced increase in myosin II activity (one h) in the lens further indicates the role of piezo channels in the regulation of myosin II activity, perhaps through decreased calcium influx. The inhibition of myosin II activity in cultured lenses treated with an inhibitor of Piezo channel activity, however, did not lead to a change in the lens tensile properties. The lenses in this study were only treated with the Piezo channel inhibitor for 48 h, and it remains to be assessed whether extended treatment with the inhibitor would result in a further decrease in myosin II activity, ultimately leading to decreased lens stiffness. Inhibition of myosin II has been reported to decrease stiffness in chicken lenses [12]. Therefore, future studies will focus on determining the role of Piezo channel activity in the regulation of myosin II activity in human lenses, which are soft and accommodative in the context of lens shape changes induced by mechanical tension and the ciliary muscle contractile force. Since Piezo1-floxed mice are readily available, the development and utilization of a Piezo1 conditional knockout mouse model would benefit future studies investigating the role of Piezo1 in lens functions. One obvious limitation of this study is that it does not address how lens shape change affects Piezo channel activity and vice versa. This aspect will be evaluated in future studies, after establishing the required experimental techniques in our laboratory.

## 4. Materials and Methods

### 4.1. Mice

All experiments using wild-type mice (C57BL/6J strain, males and females) were carried out in accordance with the recommendations of the Guide for the Care and Use of Laboratory Animals of the National Institutes of Health and the Association for Research in Vision and Ophthalmology. The animal protocol (A213-19-10) was approved by the Institutional Animal Care and Use Committee of the Duke University School of Medicine. Transgenic Piezo1-tdT (JAX stock #029214) mice were procured from the Jackson Laboratory, Bar Harbor, Maine.

### 4.2. Lens Compression Studies

Freshly dissected mouse lenses (three- to four-weeks-old, both male and female) were incubated at 37 °C under 5% CO_2_ in DMEM (Low Glucose DMEM, Invitrogen, Grand Island, NY, USA) containing penicillin (100 U/mL) and streptomycin (100 mg/mL). For the compression experiments, the lenses were submerged in 1× calcium-free phosphate-buffered saline (PBS; 2.5 mL), and a flat bottom glass stopper weighing ~3.5 g was carefully placed on the lens while supporting it to stand the stopper on the lens at room temperature. After 60 s of compression with no lens rupture, 2.5 mL of ice cold 20% trichloro acetic acid (TCA; Sigma-Aldrich, St. Louis, MO, USA) with 1 M dithiothreitol (DTT; Sigma-Aldrich) was added to the medium, and the glass stopper was removed after another 60 s. Petri dishes containing the lenses and TCA were placed on ice for an additional 4 min. The non-compressed lenses were similarly subjected to the TCA precipitation steps. Prior to homogenization, the TCA-treated lenses were imaged at room temperature using a Nikon D3000 DSLR zoom camera (Nikon Inc., Melville, NY, USA) to record and estimate compression-induced lens shape changes. The lenses were then processed to monitor changes in the levels of phosphorylated-MLC. These experiments were carried out in different batches using both the control and load-applied lenses simultaneously.

In a second set of experiments, we used glass coverslips in place of the glass stopper described above to compress the lens. For this, we used 10-mm-diameter MatTek 35 mm glass bottom micro well dishes (MatTek Corp, Ashland, MA, USA), as described by Cheng et. al. [34]. Freshly enucleated mouse lenses (3–4-weeks-old) were placed in the center of the glass bottom wells containing 2.5 mL of DMEM calcium-free medium at room temperature. Either five (1.07 g total weight) or ten (2.14 g) glass coverslips were then carefully stacked on each lens. After 60 s of compression, 2.5 mL of ice cold 20% TCA containing 1 M DTT was added to one set of the lenses. The coverslips were removed from the second set of lenses for a period of 60 s to facilitate the resilience or rebound of the lens shape. After this, ice cold TCA (20%) containing 1 M DTT was added as described above. The lenses were subsequently processed for quantitating changes in the levels of phosphorylated MLC.

Trichloroacetic-acid-precipitated lenses were washed in ice-cold deionized water for a minimum of five times, followed by one wash with diethyl ether to wash off the acid. Following this, the lenses were air dried and homogenized using a glass homogenizer in 8 M urea solution containing 20 mM Tris, 23 mM glycine, 10 mM DTT, and saturated sucrose with protease inhibitors (complete, Mini EDTA free) and PhosSTOP phosphatase inhibitors (one each/10 mL buffer; obtained from Roche, Manheim, Germany). The protein content in the 800x g lens supernatants was determined using a Micro BCA™ protein Assay Kit (Thermo Fisher Scientific, Waltham, MA, USA). Equal amounts of protein (200 μg) from the non-compressed and compressed lenses were subjected to urea/glycerol-PAGE (polyacrylamide gel electrophoresis) and Western blot analysis to detect changes in the levels of phosphorylated MLC, as we described previously [18].

### 4.3. RT-PCR and qRT-PCR

To determine the expression of Piezo channels in the lens, the total RNA was extracted from pooled neonatal (P1) and mature (P30) mouse lenses using an RNeasy Micro kit (Qiagen, Inc., Valencia, CA, USA) and reverse transcribed using the Advantage RT for PCR Kit (Clontech Laboratories, Inc., Mountain View, CA, USA). Piezo1- and Piezo2-specific DNA products were amplified using the respective forward and reverse oligonucleotide primers (Piezo-1: 5′-GCGAGCTGCTACTGGATAGG-3′/5′-GGATTCGCGGAGGGAAGTAG-3′, expected product size = 384 bp); (Piezo-2: 5′-GTCCCGCCCAATGACTACTATG-3′/5′-CCTCATCTTCCTTCGCCATCTC-3′, expected product size = 220 bp) and the Advantage^®^ 2 PCR Kit (Clontech, Mountain View, CA, USA). Briefly, gene amplification was performed using a C1000 Touch™ Thermal Cycler (Bio-Rad, Hercules, CA) with a denaturation step at 95 °C for 30 s, followed by annealing at 60 °C for 30 s and extension at 72 °C for 45 s. The cycle was repeated 30 times with a final step at 72 °C for 5 min. Amplified DNA products were separated on agarose gel electrophoresis, visualized using a Gel Red Nucleic Acid Stain (Biotium, Hayward, CA, USA), and viewed with Bio-Rad ChemiDoc™ Touch Imaging System (Bio-Rad, Hercules, CA, USA). The DNA products were excised from the gel and sequenced to confirm their identity.

For qRT-PCR, the above-prepared single-stranded cDNA libraries derived from the P30 lenses were used in the PCR master mix consisting of iQSYBR Green Supermix (Bio-Rad, Hercules, CA, USA) and the Piezo1 and Piezo2 gene-specific oligonucleotide primers described above. PCR reactions were carried out in triplicate using the following protocol: 95 °C for 2 min followed by 50 cycles of 95 °C for 15 s, 60 °C for 15 s, and 72 °C for 15 s. An extension step was used to measure the increase in the fluorescence and melting curves obtained immediately after amplification by increasing temperature in 0.4 °C increments from 65 °C for 60 cycles of 10 s each using the CFX 96™ Real-Time PCR detection system (Bio-Rad Hercules, CA, and USA). Fold differences in Piezo1 and Piezo2 gene expression in mouse lenses were normalized to glyceraldehyde 3-phosphate dehydrogenase (GAPDH) expression and calculated by the comparative threshold method, as we described previously [18].

### 4.4. Lens Organ Culture

Freshly dissected lenses from one-month-old mice were incubated at 37 °C under 5% CO2 in DMEM (Low Glucose DMEM, Invitrogen) containing penicillin (100 U/mL) and streptomycin (100 mg/mL). After acclimatization in a culture medium for 16–18 h, lenses were screened for any dissection-associated damage and elimination as we have previously described [7]. Pre-screened clear lenses were then shifted to the control media or media containing either 10 µm or 25 µm Yoda1 (Piezo agonist, Cat. No. 5586, Tocris Bioscience, Minneapolis, MN, USA) and incubated for 1, 6, or 24 h, or were incubated in media containing 2.44 µm GsMTx4 (Piezo antagonist, Cat. No. 4912, Tocris Bioscience) for 48 h. Yoda1 was dissolved in dimethyl sulfoxide (DMSO), whereas GsMTx4 was dissolved in water. The control lenses were incubated with the respective vehicles. Upon completion of the drug treatment, the lens transparency was monitored by viewing it with a dissecting light microscope. Images were then recorded under dark illumination using a Zeiss Axio Cam ERc 5 s camera (Zeiss Inc. Whiteplains, NY, USA). Lenses treated with agonist and antagonists and their respective controls were flash frozen to analyze changes in the lens membrane protein profile or processed to determine the levels of phospho-MLC and calpain activity.

### 4.5. Immunoblotting

To detect the presence of the Piezo1 protein in wild-type mouse lenses, whole-lens homogenates derived from P1, P14, P16, and lysates of the lens epithelium, and fiber mass from P21 and P27 day-old mice, were prepared. The protein solubilization buffer (prewarmed to 65 °C) containing 8 M Urea, 5% SDS, 50 mM Tris pH7.4, 5 mM EDTA, and 2.5 mM N-ethylmaleimide, along with a complete Mini, EDTA-free protease Inhibitor cocktail tablet and PhosSTOP phosphatase inhibitor cocktail tablet (1 each/10 mL buffer; obtained from Roche, Manheim, Germany), was used for homogenization. The lens lysates prepared from P1, P14, and P25 mice as described above were used for detecting the presence of the Piezo2 channel. All the fine chemicals used in this study were procured from Sigma-Aldrich, Inc. Equal amounts of protein derived from 800× *g* supernatants quantified by the Micro BCA™ protein assay kit were mixed with freshly prepared 2× SDS PAGE Laemmli buffer containing 40 mM DTT and incubated for 10 min at 65 °C. These samples were separated on 4–20% Mini-Protean^®^ TGX Stain-Free™ gradient SDS-polyacrylamide gels (Bio-Rad Hercules, CA, USA). Electrophoresis was carried out initially at 80 V for 15 min, until the samples fully entered the gel, followed by 130 V for 45 min. The proteins were electrophoretically transferred to a nitrocellulose membrane at 100 V for 1 h. The membranes were blocked with 5% blocking grade milk protein in Tris-buffered saline (TBS) buffer containing 0.1 Tween-20 (TBST), incubated overnight with a rabbit polyclonal Piezo1 antibody (1:1000 dilution, Cat No. NBP1-78537, Novus Biologicals, Centennial, CO, USA) or with a Piezo2 antibody (1:100 dilution; rabbit polyclonal, Prestige Antibody from Millipore Sigma. Cat No. HPA031975) in TBST, followed by washing and incubation with an anti-rabbit secondary antibody at a 1:4000 dilution (Thermo Fisher Scientific). The blots were developed by chemiluminescence using Chemidoc™ Touch (Bio-Rad). GAPDH was immunoblotted as a loading control using an anti-GAPDH antibody (Cat No: 60004-1, Protein tech Group, Chicago, IL, USA).

Whole-lens lysates and lens epithelium and fiber mass lysates were also prepared from transgenic Piezo1^-tdT^ mice (one-month-old) as described above. Lung tissue dissected from the same mice was processed as described above and used as a positive control [35]. Lens and lung lysates were separated on 4% SDS-PAGE and transferred to a nitrocellulose membrane at 100 V for 1 h, blocked with a 5% blocking grade milk protein in (TBST) and then incubated overnight with an anti-red fluorescent protein antibody (RFP; 1:1000, Cat. No. 600-401-379, Rockland, Inc, Limerick, PA, USA). The blots were developed as described above.

To determine the levels of total and phospho-MLC in the lens homogenates, the TCA-treated lens samples described above were subjected to immunoblot analysis using anti-phospho-MLC (pMLC) or anti-MLC rabbit polyclonal antibodies (Cat. No. 3674 and 3672, respectively, Cell Signaling Technology) as we described previously [18]. The immunoblots were developed by enhanced chemiluminescence using Chemidoc™ Touch (Bio-Rad) as described above. For all the immunoblots described, protein bands were quantified using ImageJ Software.

For the SDS-PAGE separation and Coomassie blue staining of the lens membrane proteins, the control and Yoda-1-treated mouse lenses were homogenized using a glass homogenizer in ice-cold hypotonic buffer containing 10 mM Tris buffer pH 7.4, 0.2 mM MgCl_2_, 5 mM N-ethylmaleimide, 2.0 mM Na3VO_4_, 10 mM NaF, 60 µM phenyl methyl sulfonyl fluoride, 0.4 mM iodoacetamide, and a Protease and PhosSTOP Phosphatase Inhibitor Cocktail. Homogenates were centrifuged at 800× *g* for 10 min at 4 °C, and the respective supernatants were centrifuged further at 100,000× *g* for 1 h at 4 °C using a Beckman Coulter benchtop ultracentrifuge (Optima MAX Series). The 100,000× *g* insoluble pellets were re-suspended in a hypotonic buffer and washed with same buffer, and the final membrane-enriched protein pellets were suspended in the urea sample buffer containing 8 M urea, 20 mM Tris, 23 mM glycine, 10 mM DTT, and saturated sucrose along with protease and phosphatase inhibitors. Protein concentrations were estimated (Micro BCA protein assay kit) and separated by the gradient SDS-PAGE (4–20% Criterion XT precast gels) using 1× MOPs buffer (Invitrogen) and stained overnight with a GelCode™ Blue Stain Reagent (Cat No. 24590, Thermo Fisher Scientific, Waltham, MA, USA). After destaining with deionized water, the gels were imaged using a Chemidoc™ Touch (Bio-Rad) imaging system.

### 4.6. Tissue Fixation and Immunofluorescence

Freshly enucleated non-fixed transgenic Piezo1-tdT (one-month-old) mouse lenses were embedded in optimal cutting temperature media (OCT, Tissue-Tek, Torrance, CA, USA) and flash frozen in liquid nitrogen. The lenses were cut into 10-μm-thick sections using a Microm™ HM550 Cryostat (GMI, Ramsey, MN, USA), as we previously described (6). Their cryosections were fixed in 4% paraformaldehyde for 10 min at room temperature, rinsed three times with PBS at 5 min intervals, then incubated in blocking buffer (5% globulin-free BSA and 5% filtered goat serum in 0.3% Triton X-100 containing PBS) for 30 min. The tissue sections were incubated overnight at 4 °C with anti-RFP rabbit primary antibodies (1:200 dilution in blocking buffer) in a humidified chamber. The sections were washed in 0.3% Triton-X-100 containing PBS wash buffer and incubated with Alexa-Fluor-488-conjugated secondary antibody for 1 h at room temperature (Invitrogen, Grand Island, NY, USA; 1:200 dilution). Slides were then washed 3 times in wash buffer at 10 min intervals and mounted with a VectaMount^®^ Permanent Mounting Medium (Cat No. H-5000-60, Vector Laboratories Inc, Burlingame, CA, USA). Representative immunofluorescence data were based on analysis of a minimum of three tissue sections derived from two independent specimens. Images were captured using an Eclipse 90i confocal laser-scanning microscope (Nikon Instruments, Inc., Melville, NY, USA).

### 4.7. Calpain Assay

To determine changes in the calpain activity of the Yoda-1-treated (10 µM for 24 h) versus the control lenses (DMSO-treated), the Calpain Activity Fluorometric Assay Kit was used (Cat No: K240-100, BioVision Incorporated, Milpitas, CA, USA). Briefly, the Yoda-1-treated and control lenses were homogenized (100 μL extraction buffer per lens) and incubated on ice for 20 min with gentle mixing of the samples every 5 min. The samples were centrifuged at 10,000× *g* for 60 s, and the protein was quantified in the supernatants (Micro BCA protein assay kit). Equal amounts of protein (200 μg) diluted in 85 μL of extraction buffer were transferred to the Corning^®^ 96 Well Black Polystyrene Microplate (Cat No. CLS3603, Millipore Sigma, and St. Louis, MO, USA). Control lens lysates were added to the designated wells to serve as positive and negative controls. One microliter of active calpain or the calpain inhibitor Z-LLY-FMK, provided in the kit, was added to the respective controls. A total of 10 microliters of 10× reaction buffer and 5μL calpain substrate were added to all wells and incubated at 37 °C for 1 h in the dark. Sample fluorescence was read using a SpectraMax M3 Multi-Mode Microplate Reader (Molecular Devices, LLC., San Jose, CA, USA) at a 400 nm excitation and 505 nm emission as per the manufacturer’s instructions. The calpain activity was expressed as Relative Fluorescent Units (RFU) per milligram protein of the lens homogenate.

### 4.8. Lens Stiffness Analysis

Changes in the tensile properties of the control mouse lenses (3- to 4-weeks-old) and lenses incubated with GsMTx4 (2.44 μM for 48 h) were analyzed using an RSA III micro-strain analyzer (TA Instruments, New Castle, DE, USA) equipped with parallel plate tools, as we described previously [36]. Briefly, lens compression was carried out between 2 8-mm plates, which were attached to the parallel plate tools and mounted on actuator shafts. All measurements were performed at room temperature while the lens was submerged in the culture medium. The samples were strained at a constant rate of 0.05 mm/s for a total of 35 s until sample rupture occurred. Data were acquired and plotted in real time using TA Orchestrator software. Applied stress was calculated by dividing the measured changes in the applied force by the area. The slope values before lens rupture were calculated from the linear range of the slope and plotted in Microsoft excel. The percent change in Young’s modulus between the control and GsMTx4-incubated lenses is shown as a histogram output.

### 4.9. Statistical Analyses

All data are reported as mean ± SEM (standard error of the mean) values based upon analysis of at least four independent samples, unless otherwise mentioned. GraphPad prism software (version 9.3.1; GraphPad Software, San Diego, CA, USA) was used for statistical analyses, and comparisons between two groups and three groups were performed with *t*-test and One-Way ANOVA, respectively. A *p* value of >0.05 was considered statistically significant.

## Figures and Tables

**Figure 1 ijms-23-04710-f001:**
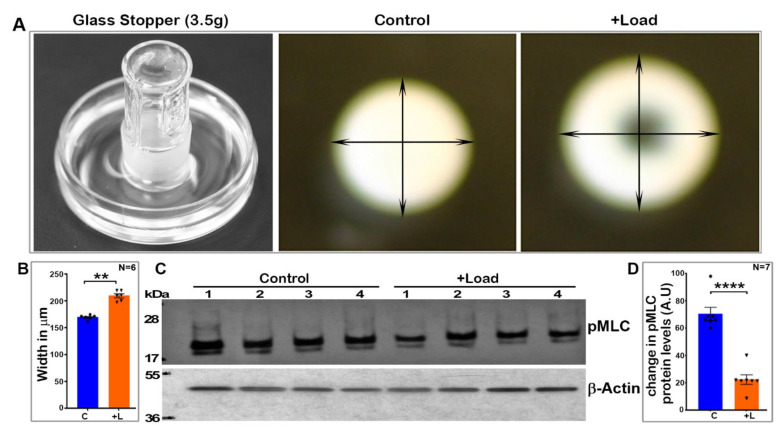
Mechanical-loading-induced lens deformation decreases intracellular myosin II activity. (**A**). To evaluate whether lens shape change induced by external mechanical loading influences myosin II activity, a glass stopper weighing ~3.5 g was placed on freshly enucleated lenses (derived from 4-week-old mice) for 60 s before fixation of the lenses with TCA as described in the Methods section. As shown in panel A (representative images), lenses subjected to mechanical loading exhibited an obvious deformation with a significant increase (by ~24%, *n* = 6) in width but no rupture, compared to control lenses (**B**,**C**). Analysis of phosphorylated MLC by immunoblotting revealed a significant decrease (>65%, *n* = 7) in the levels of phospho-MLC in the load-induced deformed lenses (lanes 1 to 4) relative to their respective controls (lanes 1 to 4) (**D**). Phospho-MLC levels were normalized to β-actin. The levels of β-actin were found to be the same between the deformed and control lenses (**C**). C: Control; L: Load-applied lenses. Values presented as Mean ± SEM. ** *p* < 0.01; **** *p* < 0.0001. A.U.: Arbitrary Units.

**Figure 2 ijms-23-04710-f002:**
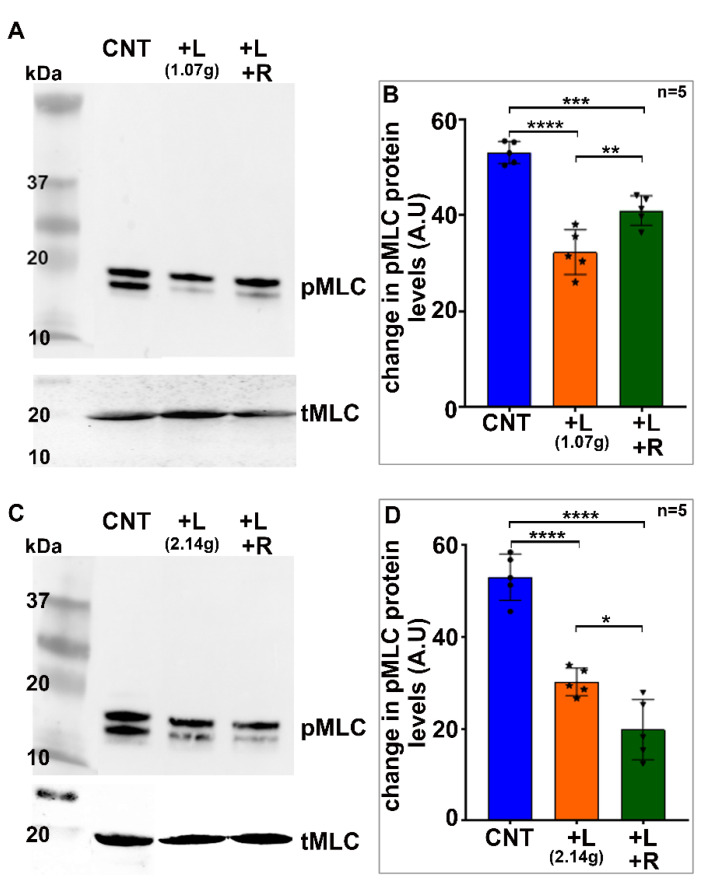
Load-induced lens deformation using glass coverslips decreases myosin II activity. For this, either 5 (total weight = 1.07 g) or 10 (total weight = 2.14 g) glass coverslips were stacked on 2 different sets of mouse lenses (from 4-week-old mice) for 60 s. Following lens compression, half of the total lenses from each set were immediately treated with TCA, while the other half were treated with TCA sixty seconds after the glass coverslips were removed, to determine the levels of phospho-MLC and total MLC. The lens-compression-induced use of glass coverslips for 60 s (**A**,**B**, 5 coverslips; **C**,**D**, 10 coverslips) was associated with an obvious decrease (ranging from 35–45%) in the levels of phospho-MLC compared to the control lenses (*n* = 5). There was a slight but significant increase (~25%) in the levels of phospho-MLC upon removal of the glass coverslips during the 60 s resilience period in the lens group loaded with 5 coverslips (**B**). Overall, however, there was still a marked decrease in the levels of phospho-MLC (by >20 and 60%, for 5- and 10-coverslip-loaded lenses, respectively) in compressed lenses allowed to ‘recover’ for 60 s after the removal of glass coverslips relative to the control lenses. Total MLC levels were not different between the compressed and uncompressed (control) lenses (**A**,**C**). Cnt: Control; L: Lenses loaded with glass coverslips for 60 s; L + R: Lenses loaded with coverslips for 60 s and allowed to ‘recover’ for 60 s prior to analysis. Values are presented as Mean ± SEM. * *p* < 0.5; ** *p* < 0.01; *** *p* < 0.001; *****p* < 0.0001. A.U.: Arbitrary Unites.

**Figure 3 ijms-23-04710-f003:**
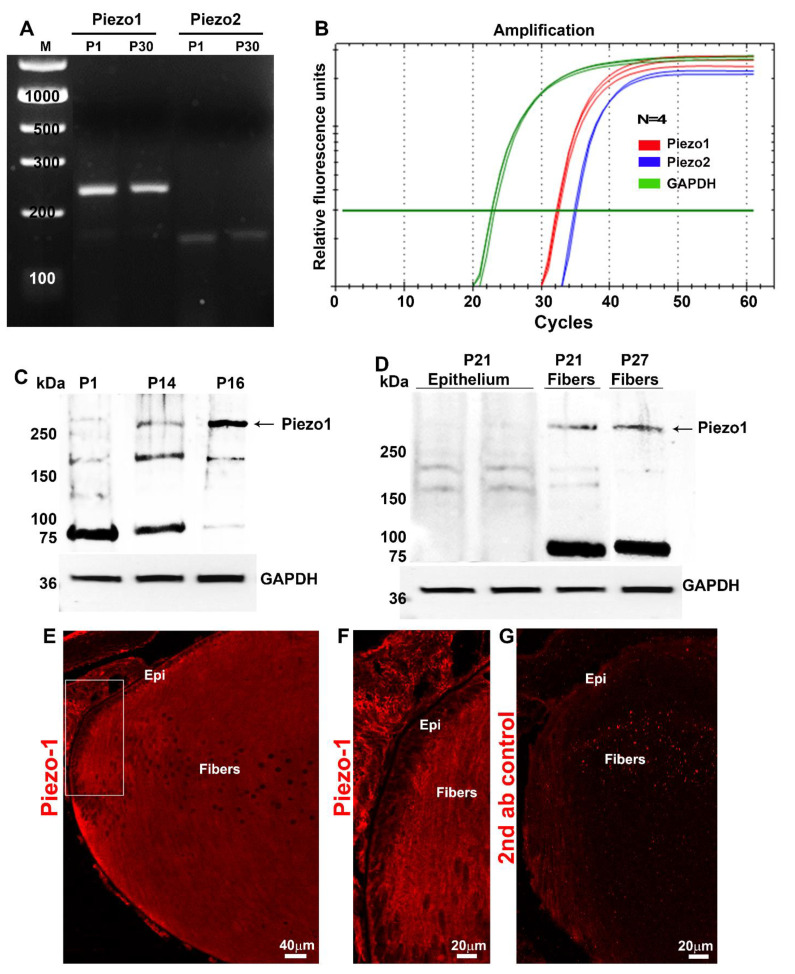
Expression and distribution of Piezo channels in mouse lenses. (**A**). RT-PCR-based confirmation of Piezo1 and Piezo2 expression in P1 and P30 mouse lenses. (**B**). qRT-PCR analysis revealed a relatively much higher level of Piezo1 expression in the lens (P30) compared to Piezo2. (**C**). Total lysates (800× *g* supernatants; 75 µg protein) derived from the P1, P14, and P16 mouse lenses analyzed using a Piezo1 polyclonal antibody exhibited immunopositive bands with an expected molecular mass of >250 kDa and >150 kDa. There was also a prominent immunopositive band at >75 kDa in the P1 and P14 lenses, the levels of which appeared to be decreased in the P16 lenses. (**D**). Piezo1 immunopositive bands of >250 and >75 kDa were present predominantly in the lens fiber samples (P21 and P27) compared to the lens epithelium (P21). (**E**,**F**). Immunofluorescence analysis of Piezo1 in the P1 mouse lens (the sagittal plane of the cryosection) revealed that the protein distributes predominantly to lens fibers relative to the epithelium (boxed area in panel (**E**) was magnified and shown in panel (**F**)). (**G**) Shows background immunofluorescence with secondary antibody alone. GAPDH: Loading control; Epi: Epithelium; Bars: Image magnification.

**Figure 4 ijms-23-04710-f004:**
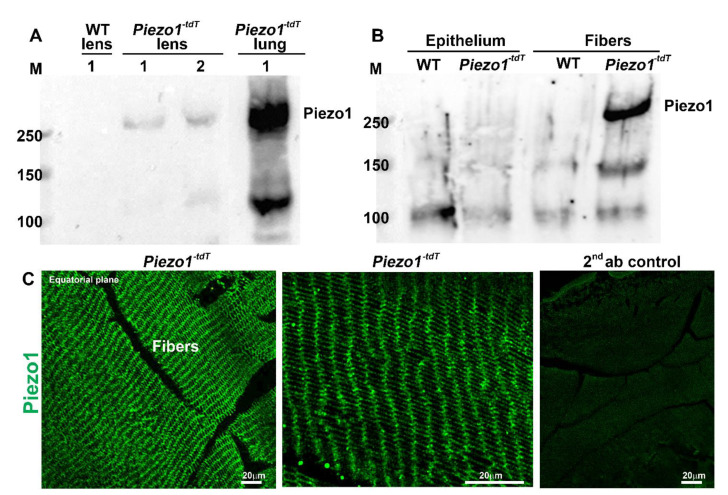
The piezo1^-tdT^ mouse confirms the expression and distribution of Piezo1 in lens fibers. (**A**). The Piezo1^-tdT^ mouse model expressing a fusion protein of Piezo1 and tdTomato (Piezo1^-tdT^) was used to determine the distribution pattern of Piezo1 in the mouse lens. Similar to what was found in the wild-type lenses (Figure 3), Piezo1^-tdT^ exhibiting the expected molecular mass of >250 kDa was detected only in P30 lens homogenates derived from the Piezo1^-tdT^ mice, but not in the wild-type lens. The positive control (lung tissue lysate from the Piezo1^-tdT^ mice) also showed a robust expression of Piezo1^-tdT^. Lanes 1 and 2 represent two different loads of the total protein (75 and 150 µg, respectively). (**B**) The Piezo1^-tdT^ fusion protein was detected predominantly in fiber cell lysates compared to lens epithelial lysates. (**C**). Immunofluorescence analysis revealed the Piezo1^-tdT^ fusion protein distributing to lens fibers with localization to both the short and long arms of the hexagonal lens fibers (Left and middle panels are with low and high magnification, respectively). The right panel shows a second antibody (Alexa Flour 488) background control fluorescence staining in the Piezo1^-tdT^ mouse lens section. Bars: Image magnification.

**Figure 5 ijms-23-04710-f005:**
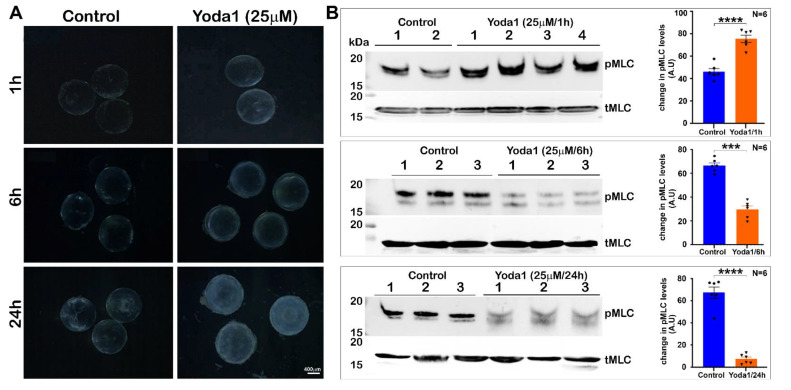
Piezo1 activation in ex vivo lenses by Yoda1: Effects on lens clarity and myosin II activity. (**A**)**.** To determine the effects of Piezo1 activation on lens transparency, lenses derived from P30 mice and maintained under culture conditions were treated with 25 µM Yoda1, an agonist of Piezo1, and changes in lens transparency were monitored for 24 h. Yoda1-treated lenses exhibited a slight haziness starting at the 1 h interval that increased progressively with time, with lenses exhibiting significant increases in swelling and a slight nuclear opacity after 24 h compared to the control lenses. (**B**). In lenses treated with Yoda1 (25 µM) for 1 h, the levels of phospho-MLC were significantly elevated compared to the untreated control lenses. On the other hand, in the 6 and 24 h Yoda1-treated lenses, there was a significant and progressive decrease in the levels of phospho-MLC (pMLC) compared to the control lenses. The levels of total MLC (tMLC), however, were found to be similar between the Yoda1-treated and control lenses throughout the course of drug treatment. Lanes 1 to 3 represent 3 experimental replicates. Bars denote image magnification. *** *p* < 0.001; **** *p* < 0.0001. A.U.: Arbitrary Units.

**Figure 6 ijms-23-04710-f006:**
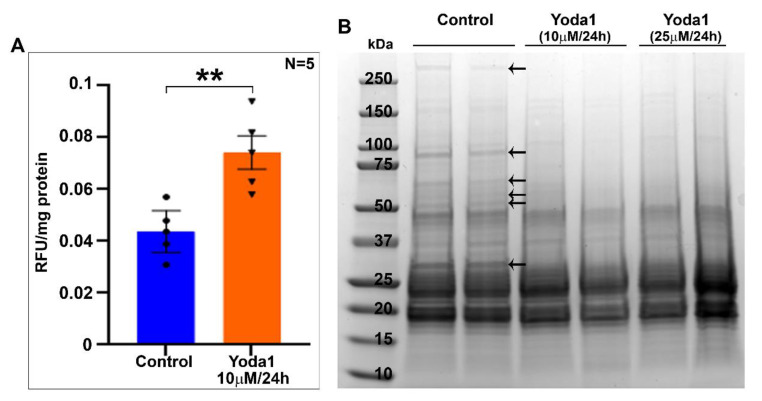
Piezo channel agonist Yoda1 treatment induces calpain activity and membrane protein proteolysis in lenses. (**A**). Calpain activity, which is calcium-dependent, was evaluated in mouse lenses treated (ex vivo) with 10 µM Yoda1 for 24 h. A significant increase (by ~69%, *n* = 5) in calpain activity was detected in the treated lenses compared to the control lenses. Values are represented as Mean ± SEM. ** *p* < 0.01. RFU: Relative fluorescence units. (**B**). Having found increased calpain activity and haziness in Yoda1-treated lenses, we then examined the integrity of lens membrane protein fractions derived from the Yoda1-treated (10 and 25 µM for 24 h) and control lenses by SDS-PAGE analysis. As shown in the figure, the Coomassie-blue-stained SDS-PAGE profile of the membrane-enriched fraction from Yoda1-treated lenses showed a markedly reduced staining of several protein bands (indicated with arrows) compared to the control lenses, indicating increased proteolysis, perhaps due to increased calcium-dependent calpain activity associated with Piezo1 activation (**A**). Data are shown for two representative lenses per group.

**Figure 7 ijms-23-04710-f007:**
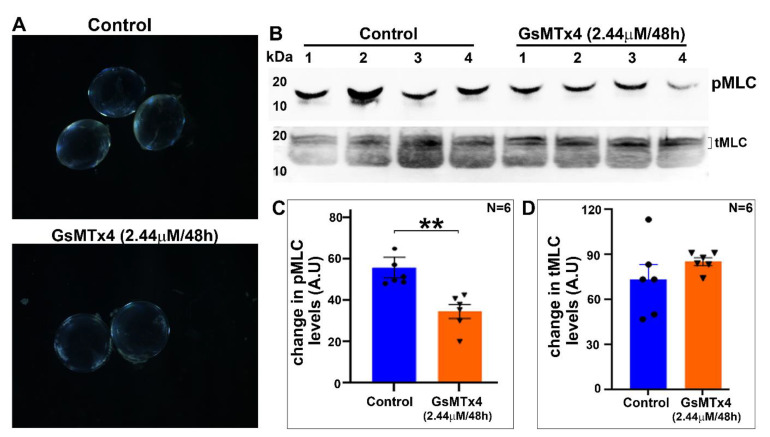
Decreased myosin II activity in Piezo1-inhibitor-treated lenses. (**A**) Lenses (from P30 mice) treated ex vivo for 48 h with 2.44 µM GsMTx4, a specific inhibitor of Piezo1, remained largely transparent, with no significant change in wet weight (4.33 ± 0.156 mg/lens, *n* = 5) compared to the control (4.14 ± 0.036 mg/lens, *n* = 5) lenses. (**B**,**C**) Relative to the control lenses, the GsMTx4-treated (48 h) lenses showed a significant (by ~38%) decrease in their levels of phospho-MLC. (**D**) The levels of total MLC in GsMTx4-treated lenses were not different relative to the control lenses (based on densitometric evaluation of the top doublet bands of tMLC shown in panel B). Values represent Mean ± SEM. ** *p* < 0.01. A.U.: Arbitrary Units.

**Figure 8 ijms-23-04710-f008:**
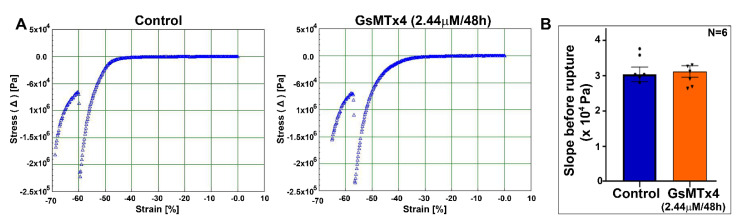
The mouse lens maintains normal tensile properties under Piezo1 channel inhibition. (**A**) A representative microstrain analyzer generated stress/strain tracings of mouse lenses from the control and GsMTx4-treated groups. (**B**) Changes in Young’s modulus calculated from the lens prerupture stress/strain slopes were not different between the GsMTx4-treated and control lenses. Values are denoted as Mean ± SEM.

## Data Availability

The raw data supporting the conclusions of this article will be made available by the authors, without undue reservation.

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
