# Peer review of "Mechanical Load and Piezo1 Channel Regulated Myosin II Activity in Mouse Lenses"

_ijms, 2022, doi:10.3390/ijms23094710_

Round 1

Reviewer 1 Report

To gain insights into the plausible involvement of mechanotransduction in lens shape change, in this study the authors have produce interesting data that shows a link between load-induced force the activation of mechano-sensitive Piezo channels I and the activity of the cytoskeletal protein myosin II. However as the authors point out the expression, distribution, and role of Piezo channels in the lens function are not well known. To fill this information void the authors have used immunoblot and immunofluorescence analyses, plus the use of a Piezo1 tdT transgenic mouse model to studythe expression and distribution of Piezo 1 in the mouse lens and how the pharmacological manipulation of Piezo 1 changes lens shape induced by load decreased myosin light chain (MLC) phosphorylation. They show that of Piezo1 by Yoda1 for one hour led to increase in the levels of phosphorylated MLC, Yoda1 treatment for an extended period led to opacification in association with increased calpain activity and degradation of membrane proteins in ex-vivo mouse lenses. In contrast, inhibition of Piezo1 by GsMTx4 decreased MLC phosphorylation but did not affect lens tensile properties. Based on this exploratory study the authors claim that changes in mechanical load are sensed by Piezo1 and transduced into changes in the regulation of myosin II activity in lens, which could alter lens shape and could be relevant to lens shape change observed during the process of accommodation.

While this is very interesting data I have a number of concerns which if addressed would strengthen this paper. Firstly, the concept that the observed changes related to the rapid and dynamic changes observed in accommodation need to be toned down. The mouse lens does not accommodate, and it is not clear whether the changes in MLC phosphorylation observed in the mouse lens would be fast enough to account for the dynamic changes in lens shape that occur in the primate lens that can accommodate. So I the first instance I would suggest the authors first claim should be that the mouse lens can actively regulate its steady state shape and of course such a shape change will the power of the lens and this may be a response to ensure that the power of the non-accommodating mouse lens can be altered as the mouse eye grows to ensure light remains focussed on the retina. Whether this process has any bearing the process of  accommodation can be raised but awaits further investigation in primate lenses that can accommodate.

My second major concern is that the authors have not really been rigorous enough with their characterisation of Piezo 1 in the lens. They identify both Piezo 1 and 2 via PCR but only characterise Piezo 1. The data presented in Western blots (Figure 3) is some what inconsistent/confusing in that it uses protein fractions from different ages of lens. The presence of the 75KDa band is not explained and it is the most prominent band. The immunolabelling of Piezo 1 is of low resolution and poor quality and is not comparable to the labelling seen in Figure 4. In figure 4 what antibody are you using to label Piezo1-tdT. Is it the same as what is used in Figure 3. If so why do you not get any signal in the wild type lens. The images in Figure 4 are very nice but they do not correlate with the Piezo 1 antibody labelling show in Figure 3 and only represent a small area of the lens that has not been defined. A full mapping of Piezo1-tdT labelling through the lens would be an excellent addition to this paper. At least images of the epithelium and underlying fiber cells is required as this is the point of this Figure to confirm that Piezo 1 expression is localised to the fiber cells.

In the organ culture experiments I found it interesting the lenses were subject to an acclimatization period of 16 to 18 hours before initiating experiments. Have the authors determined whether the base line properties of the lens change during this pre-incubation period.

Why was the incubation period different for Yoda (24 hours) and GsMTx4 (48 hours)? Also note that the legend for Figure 7 states 24 hours incubation but it is 48 hours in the text and in panel 7A. Why was the strain test applied to GsMTx4 treated lenses but not Yoda-treated lenses. 

Author Response

Ms. Corrine Gao

Dear Ms. Gao,

Sub: Response to the reviewer’s comments (Manuscript ID: ijms-1650903).

Our manuscript (Manuscript ID: ijms-1650903) entitled " Mechanical Load and Piezo1 Channel Regulated Myosin II activity in the Mouse Lens” has been revised in response to comments provided by two independent reviewers. We addressed all issues raised by the reviewers and incorporated changes (see marked copy) at appropriate places in the text and figures. We greatly appreciate the reviewers’ thorough critique and helpful suggestions, and have provided a point-by-point response to their comments below.

We included two supplemental figures (Figs. S1 and S2) in the revised manuscript to address the reviewer’s comments, and tracked all the changes made to the manuscript and legends.

We thank the reviewers for their invaluable time and constructive criticism.  We sincerely feel that this revised manuscript is now suitable for publication.

Sincerely,

  1. Vasantha Rao, Ph.D.

Common response to both the reviewers:

We started this study as an exploratory project with a Duke final year undergraduate student (Biology major) as this candidate’s required independent research project. Unfortunately, a few months into her project, due to Covid pandemic, the labs were locked down and the student went home (out of town) and never came back to the lab because after the labs were opened for a partial operation, she started her first year med school. Meanwhile, we also exhausted our resources for the lens project and did not have personnel to focus on this project. Due to these reasons, although we wished a comprehensive study, we could not perform some of the obvious experiments to link the findings of the load-induced changes directly to the Piezo channel activity. However, the data presented in this manuscript is novel and we felt that it is important to share these pilot but new findings with a broader community of vision and other researchers.        

Response to Reviewer 1:

Comments and Suggestions for Authors:

Comment 1: While this is very interesting data I have a number of concerns which if addressed would strengthen this paper. Firstly, the concept that the observed changes related to the rapid and dynamic changes observed in accommodation need to be toned down. The mouse lens does not accommodate, and it is not clear whether the changes in MLC phosphorylation observed in the mouse lens would be fast enough to account for the dynamic changes in lens shape that occur in the primate lens that can accommodate.

So the first instance I would suggest the authors first claim should be that the mouse lens can actively regulate its steady state shape and of course such a shape change will the power of the lens and this may be a response to ensure that the power of the non-accommodating mouse lens can be altered as the mouse eye grows to ensure light remains focused on the retina. Whether this process has any bearing the process of  accommodation can be raised but awaits further investigation in primate lenses that can accommodate.

Response: We thank the reviewer for this constructive suggestion and as recommended, we revised and edited the text in the Discussion section accordingly. However, it is also important to recognize that muscle contraction and relaxation (e.g. heart, lung and eyelid), and cell shape change and movement, which are regulated partly by the actomyosin contraction/relaxation are very rapid and dynamic.  

Comment 2: My second major concern is that the authors have not really been rigorous enough with their characterization of Piezo 1 in the lens. They identify both Piezo 1 and 2 via PCR but only characterize Piezo 1.

Response: In addition to the RNAseq based transcriptome profile and qRT-PCR-based quantification described for the expression of both Piezo1 & 2, we determined the presence of these proteins in the mouse lens homogenates by immunoblot analysis. While we detected the immunopositive protein bands for Piezo1, for Piezo2, there were no detectable bands. Since the data for Piezo2 were negative, we did not include these data. However, now we included Fig. S1 (Supplemental material) to show the immunoblot results for Piezo2 protein.      

Comment 3: The data presented in Western blots (Figure 3) is somewhat inconsistent/confusing in that it uses protein fractions from different ages of lens. The presence of the 75 kDa band is not explained and it is the most prominent band.

Response: The expected protein band for Piezo1 was at around 250 kDa, and in the mouse lens total homogenates, we detected a specific immunopositive band at slightly above 250 kDa. In addition to this band, we also found a prominent band at around 75 kDa and another band at above 150 kDa. Whereas in the lens fiber cell lysates, we detected both 250 and 75 kDa immunopositive protein bands but not in the lens epithelial lysates. Collectively, these results infer that lens contains the native 250 kDa Piezo1 and possibly, its proteolyticaly cleaved products. Additionally, the RT-PCR amplified results reveal a single DNA product indicating that there is most likely only one isoform for Piezo1 protein in the mouse lens. We have described in the Results section about all the Piezo1 antibody recognized proteins bands.          

Comment 4: The immunolabelling of Piezo 1 is of low resolution and poor quality and is not comparable to the labelling seen in Figure 4. In figure 4 what antibody are you using to label Piezo1-tdT. Is it the same as what is used in Figure 3. If so why do you not get any signal in the wild type lens. The images in Figure 4 are very nice but they do not correlate with the Piezo 1 antibody labelling show in Figure 3 and only represent a small area of the lens that has not been defined. A full mapping of Piezo1-tdT labelling through the lens would be an excellent addition to this paper. At least images of the epithelium and underlying fiber cells is required as this is the point of this Figure to confirm that Piezo 1 expression is localised to the fiber cells.

Response: Since we found multiple immunopositive bands for Piezo1 protein using Piezo1 antibody in the mouse lens homogenates (Fig.3), we decided to confirm the expression and distribution of Piezo1 in lens with an alternative approach. For this, we used a Piezo1-tdT transgenic mouse model in which a fusion protein of Piezo1 with the sequence for tandem-dimer Tomato (td-T, fluorescent protein) is expressed.   Since the piezo1-tdT fusion protein is expressed from the native Piezo1 promoter and regulatory elements, the levels and pattern of expression of piezo1-tdT fusion protein are expected to mimic those of the endogenous Piezo1 channel. The immunoblot and immunofluorescence data showed from the lenses of Piezo1-tdT transgenic mouse was based on using an antibody raised against the tdT reporter protein, and it is not with the Piezo1 antibody used in Fig. 3. As expected, the immunofluorescence in Fig.4 for Piezo1-tdT is much brighter and stronger compared to the data shown for the native Piezo1 channel in Fig. 3.

Secondly, the fixation and sectioning is different for the Piezo1-tdT mouse lens from the wild type lenses shown in Fig.3. For the Piezo1-tdT lenses, the tissue cryosections (from P30 lenses) were prepared prior to fixation. Therefore, we found extensive damages to the P30 lens tissue sections during the sectioning and did not get good intact (sections containing both epithelium and fibers. Whereas the data shown in Fig. 3 were from the paraffin sections which usually give the intact lens sections covering both epithelium and fiber mass. We included in the revised manuscript, the details regarding the region of the lens image showed in Fig. 4. We have now also provided a supplemental figure (Fig. S2) to show lack of Piezo1-tdT specific staining in the lens epithelium of these transgenic mice. Moreover, we purchased only very limited number of Piezo1-tdT mice to use them directly in this study and we did not breed them to develop a colony.       

Comment 5: In the organ culture experiments I found it interesting the lenses were subject to an acclimatization period of 16 to 18 hours before initiating experiments. Have the authors determined whether the base line properties of the lens change during this pre-incubation period.

Response: We followed a standard protocol for conducting the lens organ culture studies, which involves screening and elimination of the damaged lenses resulting from the dissection prior to their use. Although we did not perform any preincubation studies, generally, during the preincubation time as long as we maintain physiological osmolarity, we do not expect any overt differences in the cultured lenses from the fresh lenses.      

Comment 6: Why was the incubation period different for Yoda (24 hours) and GsMTx4 (48 hours)? Also note that the legend for Figure 7 states 24 hours incubation but it is 48 hours in the text and in panel 7A. Why was the strain test applied to GsMTx4 treated lenses but not Yoda-treated lenses. 

Response: We thank the reviewer for pointing out our inadvertent discrepancy in Fig. 7 legend and we edited the text appropriately. To our surprise, Yoda treated lenses developed opacity progressively, and within 24 hrs of treatment, there was a noticeable opacity. Therefore, we could not use them in stress/strain analysis. Whereas lenses treated with Piezo channel inhibitor (GsMTX4) stayed clear after 48 hrs, therefore, we decided to test the stiffness changes only in these lenses.

We thank the reviewer once again for his/her time and constructive comments.

Response to Reviewer 2:

Comments and Suggestions for Authors:

First, I would like to thank the authors for submitting their paper on the involvement of the Piezo 1 channel in the regulation of the activation of Mysion II activity in the mouse lens. I have several comments regarding the study.

Response: We thank the reviewer for his/her time and constructive critique to strengthen our study’s conclusions.

Comment 1: The authors presented evidence that with an applied increase of tensile force (either with glass stopper or glass coverslips) resulted in a macroscopic increase of the width of the lens. Further they showed that this leads to an increase of the pMLC and therefore increased activity of the myosin II protein. Unfortunately the authors failed to show any molecular evidence that a change in the distribution of the myosin protein takes place and identify where in the lens (i.e.outer cortex, inner cortex or core) is this happening. I would like to see a correlation experiment showing how change in activity of myosin II relates to change in structure of the lens. With other words there is no direct evidence how the change of activity of the mysion II protein is leading to structural changes in the shape of the lens that drives the increase in its width. Also, the focus is only on myosin II what about any other cytoskeletal proteins such as actin, which is the most prevalent cytoskeletal protein in the mammalian lens. Some of the REF papers referred to the chicken lens but this study uses a mouse model. Are the authors anticipating that Piezo 1 transduce changes in tension only to myosin II and to no other cytoskeletal proteins.

Response: Our general thought and hypothesis was that when lens shape is changed due to contraction and relaxation of zonules attached to the lens, there may be changes in the activities of certain channel proteins including Piezo and TRPV4 and calcium influx regulated by these channels. If this is true and if the channel protein’s activity is altered under mechanically driven  lens shape deformation, changes in the intracellular calcium is expected to influence myosin II activity, which is calcium dependent. For this, we followed myosin light chain phosphorylation, which is regulated by the calcium-dependent myosin light chain kinase activity.  In our prior published studies, we reported the effects of the inhibitors of myosin light chain kinase and L-type calcium channels on myosin light chain phosphorylation. We used in this study myosin II activity as a readout for the changes in the calcium induced intracellular activity. Moreover, if myosin II activity is altered, there will be changes in the contractile and relaxation properties of the lens and other tissues due to actomyosin driven traction force.

Since myosin light chain phosphorylation is highly dynamic, to capture the load induced effects on MCL phosphorylation, we first quickly stop all the metabolic activity as soon as we apply the load for 60 seconds using 10% TCA. The TCA fixed lenses are processed to evaluate the changes in the levels of phospho-MLC quantitatively by immunoblot analysis using the urea/glycerol gels. Since the lenses are fixed with acid, we could not follow in these lenses, the distribution profile of actin or other cytoskeletal proteins. In our published studies, we reported relative to the epithelium, fibers containing intensely distributed phospho-MLC in the mouse lenses. 

Comment 2: The transduction of Ca2+ in the lens by Piezo1 may be involved in changing the volume of the cells therefore increasing local water content in the cells. Do the authors anticipate that the AQPs are also involved in the lens width increase? 

Response: We believe that the channel proteins involved in both calcium flux and water transport could influence the lens shape change through alterations in the contractile characteristics and volume regulated force, respectively. 

Comment 3: The authors need to test the expression of Piezo 2 on a protein level despite that it has a 20x lower mRNA level expression so that we can understand if Piezo 2 is present on a protein level. 

Response: As responded to the same comment made by the reviewer one, indeed we tested for the presence of Piezo2 in the lens homogenates of mouse and we did not detect any noticeable immunopositive bands at the expected molecular mass of Piezo2 channel. We included this negative data in the revised manuscript as a supplemental material (Fig. S1).

Comment 4: I could not understand the purpose of testing the expression of Piezo1 in various postnatal stages, what is the significance or its relationship of the topic of this study provided that the majority of the experiments are carried out in P30 lenses? 

Response: The idea for the piezo channels was that since lens shape is changed due to zonnule exerted tension, whether there is an involvement for the mechanosensitive channels such as Piezo downstream to the zonnule exerted tension to induce calcium dependent activities including actomyosin organization. Towards this, since very little is known about the expression and distribution of Piezo channels in the lens, we examined for the expression and distribution profile of these channels. After confirming their expression in the lens tissue, we asked whether their activity influence calcium signaling and myosin II activity. Therefore, we examined the changes in myosin II activity under activation and inhibition of piezo channels in ex-vivo mouse lenses.  As we discussed under the limitations of our study, the next logical thing to test is whether changes in the lens shape influence the activity of piezo channels. For this aspect, first we have to learn how to monitor the changes in the piezo channel activity for which we do not have expertise and resources at present.      

Comment 5: The expression using immunolabeling approach (Fig3E) uses P1 mouse in an axial orientation. Why is this age tested. I would like to see a P30 instead.

Response: We confirmed Piezo1 expression in P27 wild type lenses (Fig. 3D) and P30 piezo1-tdT transgenic mouse lenses (Fig. 4) in addition to testing in P1 sagittal sections (Fig. 3E).

Comment 6: The Piezo1 and tdTomato model immunolabeling result (Fig4C) uses a P30 lens in an equatorial orientation (can’t be compared with Fig3E). Protein localization changes during the development therefore the same age and lens section orientation needs to be tested to compare the results. Also, Fig4C does not specify which part of the lens was imaged i.e. is this outer cortex or inner cortex and is the protein expressed in the core of the lens? 

Response: Our purpose for the distribution of Piezo1 in lens was to determine whether it is  distributed in both epithelium and fibers, and if it is distributed in the fibers, is it throughout the fibers (including both long and short arms), and in both differentiating and differentiated fibers. Therefore, we used P1 (wild type) and P30 (Piezo1-tdT transgenic). It is important to recognize that in the Piezo1-tdT mice, the expression of Piezo1 is driven by the Piezo1 promoter and regulatory mechanisms and it reflects the distribution pattern of wild type Piezo. We also provided a supplemental figure (Fig. S2) to show that as in the wild type lens (Fig. 3E), in Piezo-tdT mouse also Piezo1 is not detectable in the lens epithelium. We also included the specifics related to the region of the lens for the data shown in Fig. 4C.       

Comment 7: Effect on clarity of the lens is tested using Yoda1 or GsMTx4 after various time points and tested for change in activity of myosin II. Unfortunately this is not really relevant to your study of 60s loading tension experiments. You have not shown what happens after 60min with the activity state of myosin II with tension loading experiments and therefore is a long shot to make any correlative conclusion of the functional interrelationship between them. As an interesting observation you do show that after 1hr activation or inhibition of Piezo1 with Yoda1 or GsMTx4 does lead to an opposing effect in the activity state of MLC. What is the significance of these results please explain?

Response: We understand the reviewer’s point here, and ideally, it would have been logical to investigate whether the inhibition and activation of Piezo channel activity alters the load-induced changes in myosin II activity. This is where we encountered a surprising effect of piezo channel agonist (Yoda) on lens clarity and calpain activity. Yoda treatment started to induce opacity and activation of calpain activity in lenses. Therefore, since piezo channel inhibition did not affect lens clarity, we decided to test the changes in stiffness in these lenses. Ideally, we could have used the piezo inhibitor treated lenses and tested for the changes in lens shape by a mechanical load (glass coverslips or stopper). Again, while we were pursuing these exploratory experiments, covid pandemic restrictions have been implemented and shutting of the lab and thus, we could not finish some of the obvious experiments that we wished to perform. Now we are not in a position to take up this project work due to lack of resources and personnel. We hope to continue these and other aspects related to the lens shape change and channel protein activity in our future studies.

Comment 8: You show that change in transparency of the lens might be happening due to increase in activity of calpains in the presence of Yoda1. Interesting result but how is it fitting with your story? 

Response: Piezo channel activation has been shown to activate calcium dependent calpain activity in different cell types. In future studies it is important to determine whether piezo activation induces calpain activity in vivo or the observed results are ex-vivo lens specific. The other aspect to be considered is, whether one or more hours of Piezo activation performed in this study is non-physiological because under physiological conditions, the activation of Piezo channel may be very rapid, short, and dynamic. These aspects have to be explored in future studies and we have edited the Discussion section to include these limitations.    

Comment 9: Lastly you show that lens tensile properties remain unaffected under GsMTx4 but not under Yoda 1 presence. Why have you chosen to use one but not the other? 

Response: Unlike the GsMTx4 treated lenses, which showed no obvious change in lens clarity (after 48 hrs treatment), the Yoda treated lenses revealed activation of calpain activity and compromise in lens clarity. Therefore, we felt that the yoda treated lenses might not be ideal to use in the stress/strain analysis.

Comment 10: Is Yoda1 capable of transducing any changes to the lens afte more than 1hr what is the molecular uptake after 24hrs were the caltures changed so that the conc is maintained? 

Response: To address these aspects, we need to learn to determine the activity of piezo channel in lenses and we do not have this expertise right now.

Comment 11: Have you performed immunoprecipitation assay to identify if Piezo1 binds to myosin II protein? This may indicate if they interact physically with each other in the fiber cells.

Response:  Although we have not performed co-IP of Piezo and myosin II, what is known in the literature is that piezo channel regulates the myosin II activity through calcium influx and cell blebbing and migration. Additionally, myosin II regulated traction force (contractile force) influencing the piezo channel activity. We have cited these relevant references.  

We once again thank you for your in-depth and helpful critique and we tried our best to address all the concerns raised by you and the first reviewer.   

Reviewer 2 Report

First, I would like to thank the authors for submitting their paper on the involvement of the Piezo 1 channel in the regulation of the activation of Mysion II activity in the mouse lens. I have several comments regarding the study.

  1. The authors presented evidence that with an applied increase of tensile force (either with glass stopper or glass coverslips) resulted in a macroscopic increase of the width of the lens. Further they showed that this leads to an increase of the pMLC and therefore increased activity of the myosin II protein. Unfortunately the authors failed to show any molecular evidence that a change in the distribution of the myosin protein takes place and identify where in the lens (i.e.outer cortex, inner cortex or core) is this happening. I would like to see a correlation experiment showing how change in activity of myosin II relates to change in structure of the lens. With other words there is no direct evidence how the change of activity of the mysion II protein is leading to structural changes in the shape of the lens that drives the increase in its width. Also, the focus is only on myosin II what about any other cytoskeletal proteins such as actin, which is the most prevalent cytoskeletal protein in the mammalian lens. Some of the REF papers referred to the chicken lens but this study uses a mouse model. Are the authors anticipating that Piezo 1 transduce changes in tension only to myosin II and to no other cytoskeletal proteins.

  1. The transduction of Ca2+ in the lens by Piezo1 may be involved in changing the volume of the cells therefore increasing local water content in the cells. Do the authors anticipate that the AQPs are also involved in the lens width increase?

  1. The authors need to test the expression of Piezo 2 on a protein level despite that it has a 20x lower mRNA level expression so that we can understand if Piezo 2 is present on a protein level.

  1. I could not understand the purpose of testing the expression of Piezo1 in various postnatal stages, what is the significance or its relationship of the topic of this study provided that the majority of the experiments are carried out in P30 lenses?

  1. The expression using immunolabeling approach (Fig3E) uses P1 mouse in an axial orientation. Why is this age tested. I would like to see a P30 instead.

  1. The Piezo1 and tdTomato model immunolabeling result (Fig4C) uses a P30 lens in an equatorial orientation (can’t be compared with Fig3E). Protein localization changes during the development therefore the same age and lens section orientation needs to be tested to compare the results. Also, Fig4C does not specify which part of the lens was imaged i.e. is this outer cortex or inner cortex and is the protein expressed in the core of the lens?

  1. Effect on clarity of the lens is tested using Yoda1 or GsMTx4 after various time points and tested for change in activity of myosin II. Unfortunately this is not really relevant to your study of 60s loading tension experiments. You have not shown what happens after 60min with the activity state of myosin II with tension loading experiments and therefore is a long shot to make any correlative conclusion of the functional interrelationship between them. As an interesting observation you do show that after 1hr activation or inhibition of Piezo1 with Yoda1 or GsMTx4 does lead to an opposing effect in the activity state of MLC. What is the significance of these results please explain?

  1. You show that change in transparency of the lens might be happening due to increase in activity of calpains in the presence of Yoda1. Interesting result but how is it fitting with your story?

  1. Lastly you show that lens tensile properties remain unaffected under GsMTx4 but not under Yoda 1 presence. Why have you chosen to use one but not the other?

  1. Is Yoda1 capable of transducing any changes to the lens afte more than 1hr what is the molecular uptake after 24hrs were the caltures changed so that the conc is maintained?

  1. Have you performed immunoprecipitation assay to identify if Piezo1 binds to myosin II protein? This may indicate if they interact physically with each other in the fiber cells.

Author Response

Ms. Corrine Gao

Dear Ms. Gao,

Sub: Response to the reviewer’s comments (Manuscript ID: ijms-1650903).

Our manuscript (Manuscript ID: ijms-1650903) entitled " Mechanical Load and Piezo1 Channel Regulated Myosin II activity in the Mouse Lens” has been revised in response to comments provided by two independent reviewers. We addressed all issues raised by the reviewers and incorporated changes (see marked copy) at appropriate places in the text and figures. We greatly appreciate the reviewers’ thorough critique and helpful suggestions, and have provided a point-by-point response to their comments below.

We included two supplemental figures (Figs. S1 and S2) in the revised manuscript to address the reviewer’s comments, and tracked all the changes made to the manuscript and legends.

We thank the reviewers for their invaluable time and constructive criticism.  We sincerely feel that this revised manuscript is now suitable for publication.

Sincerely,

Vasantha Rao, Ph.D.

Common response to both the reviewers:

 We started this study as an exploratory project with a Duke final year undergraduate student (Biology major) as this candidate’s required independent research project. Unfortunately, a few months into her project, due to Covid pandemic, the labs were locked down and the student went home (out of town) and never came back to the lab because after the labs were opened for a partial operation, she started her first year med school. Meanwhile, we also exhausted our resources for the lens project and did not have personnel to focus on this project. Due to these reasons, although we wished a comprehensive study, we could not perform some of the obvious experiments to link the findings of the load-induced changes directly to the Piezo channel activity. However, the data presented in this manuscript is novel and we felt that it is important to share these pilot but new findings with a broader community of vision and other researchers.        

 Response to Reviewer 1:
Comments and Suggestions for Authors:

Comment 1: While this is very interesting data I have a number of concerns which if addressed would strengthen this paper. Firstly, the concept that the observed changes related to the rapid and dynamic changes observed in accommodation need to be toned down. The mouse lens does not accommodate, and it is not clear whether the changes in MLC phosphorylation observed in the mouse lens would be fast enough to account for the dynamic changes in lens shape that occur in the primate lens that can accommodate.

So the first instance I would suggest the authors first claim should be that the mouse lens can actively regulate its steady state shape and of course such a shape change will the power of the lens and this may be a response to ensure that the power of the non-accommodating mouse lens can be altered as the mouse eye grows to ensure light remains focused on the retina. Whether this process has any bearing the process of  accommodation can be raised but awaits further investigation in primate lenses that can accommodate.

Response: We thank the reviewer for this constructive suggestion and as recommended, we revised and edited the text in the Discussion section accordingly. However, it is also important to recognize that muscle contraction and relaxation (e.g. heart, lung and eyelid), and cell shape change and movement, which are regulated partly by the actomyosin contraction/relaxation are very rapid and dynamic.  

Comment 2: My second major concern is that the authors have not really been rigorous enough with their characterization of Piezo 1 in the lens. They identify both Piezo 1 and 2 via PCR but only characterize Piezo 1.

Response: In addition to the RNAseq based transcriptome profile and qRT-PCR-based quantification described for the expression of both Piezo1 & 2, we determined the presence of these proteins in the mouse lens homogenates by immunoblot analysis. While we detected the immunopositive protein bands for Piezo1, for Piezo2, there were no detectable bands. Since the data for Piezo2 were negative, we did not include these data. However, now we included Fig. S1 (Supplemental material) to show the immunoblot results for Piezo2 protein.      

Comment 3: The data presented in Western blots (Figure 3) is somewhat inconsistent/confusing in that it uses protein fractions from different ages of lens. The presence of the 75 kDa band is not explained and it is the most prominent band.

Response: The expected protein band for Piezo1 was at around 250 kDa, and in the mouse lens total homogenates, we detected a specific immunopositive band at slightly above 250 kDa. In addition to this band, we also found a prominent band at around 75 kDa and another band at above 150 kDa. Whereas in the lens fiber cell lysates, we detected both 250 and 75 kDa immunopositive protein bands but not in the lens epithelial lysates. Collectively, these results infer that lens contains the native 250 kDa Piezo1 and possibly, its proteolyticaly cleaved products. Additionally, the RT-PCR amplified results reveal a single DNA product indicating that there is most likely only one isoform for Piezo1 protein in the mouse lens. We have described in the Results section about all the Piezo1 antibody recognized proteins bands.          

Comment 4: The immunolabelling of Piezo 1 is of low resolution and poor quality and is not comparable to the labelling seen in Figure 4. In figure 4 what antibody are you using to label Piezo1-tdT. Is it the same as what is used in Figure 3. If so why do you not get any signal in the wild type lens. The images in Figure 4 are very nice but they do not correlate with the Piezo 1 antibody labelling show in Figure 3 and only represent a small area of the lens that has not been defined. A full mapping of Piezo1-tdT labelling through the lens would be an excellent addition to this paper. At least images of the epithelium and underlying fiber cells is required as this is the point of this Figure to confirm that Piezo 1 expression is localised to the fiber cells.

Response: Since we found multiple immunopositive bands for Piezo1 protein using Piezo1 antibody in the mouse lens homogenates (Fig.3), we decided to confirm the expression and distribution of Piezo1 in lens with an alternative approach. For this, we used a Piezo1-tdT transgenic mouse model in which a fusion protein of Piezo1 with the sequence for tandem-dimer Tomato (td-T, fluorescent protein) is expressed.   Since the piezo1-tdT fusion protein is expressed from the native Piezo1 promoter and regulatory elements, the levels and pattern of expression of piezo1-tdT fusion protein are expected to mimic those of the endogenous Piezo1 channel. The immunoblot and immunofluorescence data showed from the lenses of Piezo1-tdT transgenic mouse was based on using an antibody raised against the tdT reporter protein, and it is not with the Piezo1 antibody used in Fig. 3. As expected, the immunofluorescence in Fig.4 for Piezo1-tdT is much brighter and stronger compared to the data shown for the native Piezo1 channel in Fig. 3.

Secondly, the fixation and sectioning is different for the Piezo1-tdT mouse lens from the wild type lenses shown in Fig.3. For the Piezo1-tdT lenses, the tissue cryosections (from P30 lenses) were prepared prior to fixation. Therefore, we found extensive damages to the P30 lens tissue sections during the sectioning and did not get good intact (sections containing both epithelium and fibers. Whereas the data shown in Fig. 3 were from the paraffin sections which usually give the intact lens sections covering both epithelium and fiber mass. We included in the revised manuscript, the details regarding the region of the lens image showed in Fig. 4. We have now also provided a supplemental figure (Fig. S2) to show lack of Piezo1-tdT specific staining in the lens epithelium of these transgenic mice. Moreover, we purchased only very limited number of Piezo1-tdT mice to use them directly in this study and we did not breed them to develop a colony.       

Comment 5: In the organ culture experiments I found it interesting the lenses were subject to an acclimatization period of 16 to 18 hours before initiating experiments. Have the authors determined whether the base line properties of the lens change during this pre-incubation period.

Response: We followed a standard protocol for conducting the lens organ culture studies, which involves screening and elimination of the damaged lenses resulting from the dissection prior to their use. Although we did not perform any preincubation studies, generally, during the preincubation time as long as we maintain physiological osmolarity, we do not expect any overt differences in the cultured lenses from the fresh lenses.      

Comment 6: Why was the incubation period different for Yoda (24 hours) and GsMTx4 (48 hours)? Also note that the legend for Figure 7 states 24 hours incubation but it is 48 hours in the text and in panel 7A. Why was the strain test applied to GsMTx4 treated lenses but not Yoda-treated lenses. 

Response: We thank the reviewer for pointing out our inadvertent discrepancy in Fig. 7 legend and we edited the text appropriately. To our surprise, Yoda treated lenses developed opacity progressively, and within 24 hrs of treatment, there was a noticeable opacity. Therefore, we could not use them in stress/strain analysis. Whereas lenses treated with Piezo channel inhibitor (GsMTX4) stayed clear after 48 hrs, therefore, we decided to test the stiffness changes only in these lenses.

We thank the reviewer once again for his/her time and constructive comments.

Response to Reviewer 2:

 Comments and Suggestions for Authors:

First, I would like to thank the authors for submitting their paper on the involvement of the Piezo 1 channel in the regulation of the activation of Mysion II activity in the mouse lens. I have several comments regarding the study.

Response: We thank the reviewer for his/her time and constructive critique to strengthen our study’s conclusions.

Comment 1: The authors presented evidence that with an applied increase of tensile force (either with glass stopper or glass coverslips) resulted in a macroscopic increase of the width of the lens. Further they showed that this leads to an increase of the pMLC and therefore increased activity of the myosin II protein. Unfortunately the authors failed to show any molecular evidence that a change in the distribution of the myosin protein takes place and identify where in the lens (i.e.outer cortex, inner cortex or core) is this happening. I would like to see a correlation experiment showing how change in activity of myosin II relates to change in structure of the lens. With other words there is no direct evidence how the change of activity of the mysion II protein is leading to structural changes in the shape of the lens that drives the increase in its width. Also, the focus is only on myosin II what about any other cytoskeletal proteins such as actin, which is the most prevalent cytoskeletal protein in the mammalian lens. Some of the REF papers referred to the chicken lens but this study uses a mouse model. Are the authors anticipating that Piezo 1 transduce changes in tension only to myosin II and to no other cytoskeletal proteins.

Response: Our general thought and hypothesis was that when lens shape is changed due to contraction and relaxation of zonules attached to the lens, there may be changes in the activities of certain channel proteins including Piezo and TRPV4 and calcium influx regulated by these channels. If this is true and if the channel protein’s activity is altered under mechanically driven  lens shape deformation, changes in the intracellular calcium is expected to influence myosin II activity, which is calcium dependent. For this, we followed myosin light chain phosphorylation, which is regulated by the calcium-dependent myosin light chain kinase activity.  In our prior published studies, we reported the effects of the inhibitors of myosin light chain kinase and L-type calcium channels on myosin light chain phosphorylation. We used in this study myosin II activity as a readout for the changes in the calcium induced intracellular activity. Moreover, if myosin II activity is altered, there will be changes in the contractile and relaxation properties of the lens and other tissues due to actomyosin driven traction force.

Since myosin light chain phosphorylation is highly dynamic, to capture the load induced effects on MCL phosphorylation, we first quickly stop all the metabolic activity as soon as we apply the load for 60 seconds using 10% TCA. The TCA fixed lenses are processed to evaluate the changes in the levels of phospho-MLC quantitatively by immunoblot analysis using the urea/glycerol gels. Since the lenses are fixed with acid, we could not follow in these lenses, the distribution profile of actin or other cytoskeletal proteins. In our published studies, we reported relative to the epithelium, fibers containing intensely distributed phospho-MLC in the mouse lenses. 

Comment 2: The transduction of Ca2+ in the lens by Piezo1 may be involved in changing the volume of the cells therefore increasing local water content in the cells. Do the authors anticipate that the AQPs are also involved in the lens width increase? 

Response: We believe that the channel proteins involved in both calcium flux and water transport could influence the lens shape change through alterations in the contractile characteristics and volume regulated force, respectively. 

Comment 3: The authors need to test the expression of Piezo 2 on a protein level despite that it has a 20x lower mRNA level expression so that we can understand if Piezo 2 is present on a protein level. 

Response: As responded to the same comment made by the reviewer one, indeed we tested for the presence of Piezo2 in the lens homogenates of mouse and we did not detect any noticeable immunopositive bands at the expected molecular mass of Piezo2 channel. We included this negative data in the revised manuscript as a supplemental material (Fig. S1).

Comment 4: I could not understand the purpose of testing the expression of Piezo1 in various postnatal stages, what is the significance or its relationship of the topic of this study provided that the majority of the experiments are carried out in P30 lenses? 

Response: The idea for the piezo channels was that since lens shape is changed due to zonnule exerted tension, whether there is an involvement for the mechanosensitive channels such as Piezo downstream to the zonnule exerted tension to induce calcium dependent activities including actomyosin organization. Towards this, since very little is known about the expression and distribution of Piezo channels in the lens, we examined for the expression and distribution profile of these channels. After confirming their expression in the lens tissue, we asked whether their activity influence calcium signaling and myosin II activity. Therefore, we examined the changes in myosin II activity under activation and inhibition of piezo channels in ex-vivo mouse lenses.  As we discussed under the limitations of our study, the next logical thing to test is whether changes in the lens shape influence the activity of piezo channels. For this aspect, first we have to learn how to monitor the changes in the piezo channel activity for which we do not have expertise and resources at present.      

Comment 5: The expression using immunolabeling approach (Fig3E) uses P1 mouse in an axial orientation. Why is this age tested. I would like to see a P30 instead.

Response: We confirmed Piezo1 expression in P27 wild type lenses (Fig. 3D) and P30 piezo1-tdT transgenic mouse lenses (Fig. 4) in addition to testing in P1 sagittal sections (Fig. 3E).

Comment 6: The Piezo1 and tdTomato model immunolabeling result (Fig4C) uses a P30 lens in an equatorial orientation (can’t be compared with Fig3E). Protein localization changes during the development therefore the same age and lens section orientation needs to be tested to compare the results. Also, Fig4C does not specify which part of the lens was imaged i.e. is this outer cortex or inner cortex and is the protein expressed in the core of the lens? 

Response: Our purpose for the distribution of Piezo1 in lens was to determine whether it is  distributed in both epithelium and fibers, and if it is distributed in the fibers, is it throughout the fibers (including both long and short arms), and in both differentiating and differentiated fibers. Therefore, we used P1 (wild type) and P30 (Piezo1-tdT transgenic). It is important to recognize that in the Piezo1-tdT mice, the expression of Piezo1 is driven by the Piezo1 promoter and regulatory mechanisms and it reflects the distribution pattern of wild type Piezo. We also provided a supplemental figure (Fig. S2) to show that as in the wild type lens (Fig. 3E), in Piezo-tdT mouse also Piezo1 is not detectable in the lens epithelium. We also included the specifics related to the region of the lens for the data shown in Fig. 4C.       

Comment 7: Effect on clarity of the lens is tested using Yoda1 or GsMTx4 after various time points and tested for change in activity of myosin II. Unfortunately this is not really relevant to your study of 60s loading tension experiments. You have not shown what happens after 60min with the activity state of myosin II with tension loading experiments and therefore is a long shot to make any correlative conclusion of the functional interrelationship between them. As an interesting observation you do show that after 1hr activation or inhibition of Piezo1 with Yoda1 or GsMTx4 does lead to an opposing effect in the activity state of MLC. What is the significance of these results please explain?

Response: We understand the reviewer’s point here, and ideally, it would have been logical to investigate whether the inhibition and activation of Piezo channel activity alters the load-induced changes in myosin II activity. This is where we encountered a surprising effect of piezo channel agonist (Yoda) on lens clarity and calpain activity. Yoda treatment started to induce opacity and activation of calpain activity in lenses. Therefore, since piezo channel inhibition did not affect lens clarity, we decided to test the changes in stiffness in these lenses. Ideally, we could have used the piezo inhibitor treated lenses and tested for the changes in lens shape by a mechanical load (glass coverslips or stopper). Again, while we were pursuing these exploratory experiments, covid pandemic restrictions have been implemented and shutting of the lab and thus, we could not finish some of the obvious experiments that we wished to perform. Now we are not in a position to take up this project work due to lack of resources and personnel. We hope to continue these and other aspects related to the lens shape change and channel protein activity in our future studies.

Comment 8: You show that change in transparency of the lens might be happening due to increase in activity of calpains in the presence of Yoda1. Interesting result but how is it fitting with your story? 

Response: Piezo channel activation has been shown to activate calcium dependent calpain activity in different cell types. In future studies it is important to determine whether piezo activation induces calpain activity in vivo or the observed results are ex-vivo lens specific. The other aspect to be considered is, whether one or more hours of Piezo activation performed in this study is non-physiological because under physiological conditions, the activation of Piezo channel may be very rapid, short, and dynamic. These aspects have to be explored in future studies and we have edited the Discussion section to include these limitations.    

Comment 9: Lastly you show that lens tensile properties remain unaffected under GsMTx4 but not under Yoda 1 presence. Why have you chosen to use one but not the other? 

Response: Unlike the GsMTx4 treated lenses, which showed no obvious change in lens clarity (after 48 hrs treatment), the Yoda treated lenses revealed activation of calpain activity and compromise in lens clarity. Therefore, we felt that the yoda treated lenses might not be ideal to use in the stress/strain analysis.

Comment 10: Is Yoda1 capable of transducing any changes to the lens afte more than 1hr what is the molecular uptake after 24hrs were the caltures changed so that the conc is maintained? 

Response: To address these aspects, we need to learn to determine the activity of piezo channel in lenses and we do not have this expertise right now.

Comment 11: Have you performed immunoprecipitation assay to identify if Piezo1 binds to myosin II protein? This may indicate if they interact physically with each other in the fiber cells.

Response:  Although we have not performed co-IP of Piezo and myosin II, what is known in the literature is that piezo channel regulates the myosin II activity through calcium influx and cell blebbing and migration. Additionally, myosin II regulated traction force (contractile force) influencing the piezo channel activity. We have cited these relevant references.  

We once again thank you for your in-depth and helpful critique and we tried our best to address all the concerns raised by you and the first reviewer.   

Round 2

Reviewer 1 Report

Comments and Suggestions for Authors:

Comment 1: While this is very interesting data I have a number of concerns which if addressed would strengthen this paper. Firstly, the concept that the observed changes related to the rapid and dynamic changes observed in accommodation need to be toned down. The mouse lens does not accommodate, and it is not clear whether the changes in MLC phosphorylation observed in the mouse lens would be fast enough to account for the dynamic changes in lens shape that occur in the primate lens that can accommodate. So the first instance I would suggest the authors first claim should be that the mouse lens can actively regulate its steady state shape and of course such a shape change will the power of the lens and this may be a response to ensure that the power of the non-accommodating mouse lens can be altered as the mouse eye grows to ensure light remains focused on the retina. Whether this process has any bearing the process of accommodation can be raised but awaits further investigation in primate lenses that can accommodate.

Response: We thank the reviewer for this constructive suggestion and as recommended, we revised and edited the text in the Discussion section accordingly. However, it is also important to recognize that muscle contraction and relaxation (e.g. heart, lung and eyelid), and cell shape change and movement, which are regulated partly by the actomyosin contraction/relaxation are very rapid and dynamic.

Reply to response: I think this revision can be strengthen future

Comment 2: My second major concern is that the authors have not really been rigorous enough with their characterization of Piezo 1 in the lens. They identify both Piezo 1 and 2 via PCR but only characterize Piezo 1.

Response: In addition to the RNAseq based transcriptome profile and qRT-PCR-based quantification described for the expression of both Piezo1 & 2, we determined the presence of these proteins in the mouse lens homogenates by immunoblot analysis. While we detected the immunopositive protein bands for Piezo1, for Piezo2, there were no detectable bands. Since the data for Piezo2 were negative, we did not include these data. However, now we included Fig. S1 (Supplemental material) to show the immunoblot results for Piezo2 protein.

Reply to response: This is acceptable

Comment 3: The data presented in Western blots (Figure 3) is somewhat inconsistent/confusing in that it uses protein fractions from different ages of lens. The presence of the 75 kDa band is not explained and it is the most prominent band.

Response: The expected protein band for Piezo1 was at around 250 kDa, and in the mouse lens total homogenates, we detected a specific immunopositive band at slightly above 250 kDa. In addition to this band, we also found a prominent band at around 75 kDa and another band at above 150 kDa. Whereas in the lens fiber cell lysates, we detected both 250 and 75 kDa immunopositive protein bands but not in the lens epithelial lysates. Collectively, these results infer that lens contains the native 250 kDa Piezo1 and possibly, its proteolyticaly cleaved products. Additionally, the RTPCR amplified results reveal a single DNA product indicating that there is most likely only one isoform for Piezo1 protein in the mouse lens. We have described in the Results section about all the Piezo1 antibody recognized proteins bands.

Reply to response: I didn’t see in the results where the 75 kDa band was described as a degradation product. I still think these bands need to be described in more detail. Do the authors have  labelling from a control tissue for Piezo 1?

Comment 4: The immunolabelling of Piezo 1 is of low resolution and poor quality and is not comparable to the labelling seen in Figure 4. In figure 4 what antibody are you using to label Piezo1-tdT. Is it the same as what is used in Figure 3. If so why do you not get any signal in the wild type lens. The images in Figure 4 are very nice but they do not correlate with the Piezo 1 antibody labelling show in Figure 3 and only represent a small area of the lens that has not been defined. A full mapping of Piezo1-tdT labelling through the lens would be an excellent addition to this paper. At least images of the epithelium and underlying fiber cells is required as this is the point of this Figure to confirm that Piezo 1 expression is localised to the fiber cells.

Response: Since we found multiple immunopositive bands for Piezo1 protein using Piezo1 antibody in the mouse lens homogenates (Fig.3), we decided to confirm the expression and distribution of Piezo1 in lens with an alternative approach. For this, we used a Piezo1-tdT transgenic mouse model in which a fusion protein of Piezo1 with the sequence for tandem-dimer Tomato (td-T, fluorescent protein) is expressed. Since the piezo1-tdT fusion protein is expressed from the native Piezo1 promoter and regulatory elements, the levels and pattern of expression of piezo1-tdT fusion protein are expected to mimic those of the endogenous Piezo1 channel. The immunoblot and immunofluorescence data showed from the lenses of Piezo1-tdT transgenic mouse was based on using an antibody raised against the tdT reporter protein, and it is not with the Piezo1 antibody used in Fig. 3. As expected, the immunofluorescence in Fig.4 for Piezo1-tdT is much brighter and stronger compared to the data shown for the native Piezo1 channel in Fig. 3. Secondly, the fixation and sectioning is different for the Piezo1-tdT mouse lens from the wild type lenses shown in Fig.3. For the Piezo1-tdT lenses, the tissue cryosections (from P30 lenses) were prepared prior to fixation. Therefore, we found extensive damages to the P30 lens tissue sections during the sectioning and did not get good intact (sections containing both epithelium and fibers. Whereas the data shown in Fig. 3 were from the paraffin sections which usually give the intact lens sections covering both epithelium and fiber mass. We included in the revised manuscript, the details regarding the region of the lens image showed in Fig. 4. We have now also provided a supplemental figure (Fig. S2) to show lack of Piezo1-tdT specific staining in the lens epithelium of these transgenic mice. Moreover, we purchased only very limited number of Piezo1-tdT mice to use them directly in this study and we did not breed them to develop a colony.

Reply to response: Sorry I am still not satisfy with the quality of the immunolabelling. It is difficult to tell in Figure S2 if the epithelium is actually attached to the lens. Did you label for nuclei? The lack of more transgenic mice means that these experiments will be difficult to repeat to get improve the sections and get the distribution of Piezo 1 required I would suggest that the immune labelling shown in both Figure 3 and 4 be excluded from the paper and the authors concentration more on their western blotting data.

 Comment 5: In the organ culture experiments I found it interesting the lenses were subject to an acclimatization period of 16 to 18 hours before initiating experiments. Have the authors determined whether the base line properties of the lens change during this pre-incubation period.

Response: We followed a standard protocol for conducting the lens organ culture studies, which involves screening and elimination of the damaged lenses resulting from the dissection prior to their use. Although we did not perform any preincubation studies, generally, during the preincubation time as long as we maintain physiological osmolarity, we do not expect any overt differences in the cultured lenses from the fresh lenses.

Reply to response: This is at odds with a recent publication from Nakazawa et al that showed the distribution of another mechano-sensitive channel was altered by cutting the lens zonules and organ culturing lenses. This should be mentioned as a possibility

Comment 6: Why was the incubation period different for Yoda (24 hours) and GsMTx4 (48 hours)? Also note that the legend for Figure 7 states 24 hours incubation but it is 48 hours in the text and in panel 7A. Why was the strain test applied to GsMTx4 treated lenses but not Yodatreated lenses.

Response: We thank the reviewer for pointing out our inadvertent discrepancy in Fig. 7 legend and we edited the text appropriately. To our surprise, Yoda treated lenses developed opacity progressively, and within 24 hrs of treatment, there was a noticeable opacity. Therefore, we could not use them in stress/strain analysis. Whereas lenses treated with Piezo channel inhibitor (GsMTX4) stayed clear after 48 hrs, therefore, we decided to test the stiffness changes only in these lenses. We thank the reviewer once again for his/her time and constructive comments.

Reply to response: OK

Author Response

Response to Reviewer 1:

Comment 1: The concept that the observed changes related to the rapid and dynamic changes observed in accommodation need to be toned down.

Comment to initial response: I think this revision can be strengthen future

Response: We have discussed the limitation of mouse lens model used in the described studies, and the investigated load effects on myosin II activity especially during resilience would be different between the non-accommodating (hard) verses accommodating (soft) lenses. We were very cautious in extending our mouse lens findings directly to the accommodating lenses. We also discussed the limitations and possible differences between the ex-vivo and in vivo lens experiments. The Discussion section was revised accordingly. Our focus in this study was on the regulation of myosin II activity.     

Comment 3: The data presented in Western blots (Figure 3) is somewhat inconsistent/confusing in that it uses protein fractions from different ages of lens. The presence of the 75 kDa band is not explained and it is the most prominent band.

Comment to initial response: I didn’t see in the results where the 75 kDa band was described as a degradation product. I still think these bands need to be described in more detail. Do the authors have  labelling from a control tissue for Piezo 1?

Response: In the revised manuscript, we discussed in the Discussion section about the possible proteolytic posttranslational changes of Piezo1 in the lens fibers. Additionally, we also revised Fig. 3 to include a control for showing the background staining of secondary antibody (Fig. 3G) used in evaluating the distribution profile of Piezo1 in mouse lens (Fig. 3E &F). As can be seen from the control, there is a strong and specific positive staining with Piezo1 antibody compared to the non-specific background staining.     

Comment 4: The immunolabelling of Piezo 1

Comment to initial response: Sorry I am still not satisfy with the quality of the immunolabelling. It is difficult to tell in Figure S2 if the epithelium is actually attached to the lens. Did you label for nuclei? The lack of more transgenic mice means that these experiments will be difficult to repeat to get improve the sections and get the distribution of Piezo 1 required I would suggest that the immune labelling shown in both Figure 3 and 4 be excluded from the paper and the authors concentration more on their western blotting data.

Response: Regarding the differential distribution of Piezo1 described in the lens epithelium verses lens fibers, we supported the findings based on the results derived from the immunoblot, immunofluorescence (using wild type lens) and Piezo1-tdT mouse lens analysis. As we responded previously, the lens sections derived from the Piezo1-tdT were not from the paraffin-embedded lens sections, they were cut prior to fixation with cryo microtome and they were from the one month-old mice and had fractures. In Fig. S2, although, the epithelium was not attached to the fibers continuously, one can also see clearly the area where the epithelium was attached to the fibers (indicated with arrows) of the specimen. Unfortunately, we did not stain these tissue sections for the nuclei. Overall, we are not only convinced with our results but also invested so much effort and resources, therefore, we sincerely feel that the data shown in Figs. 3 & 4, support our conclusions in describing the distribution pattern of Piezo1 in mouse lens. Therefore, we wish to include the immunofluorescence data derived from both wild type and Piezo1-tdT transgenic mouse lenses.      

Comment to initial response: This is at odds with a recent publication from Nakazawa et al that showed the distribution of another mechano-sensitive channel was altered by cutting the lens zonules and organ culturing lenses. This should be mentioned as a possibility.

Response: We thank the reviewer for recommending the above referred reference. In the revised manuscript, we not only cited this new reference but also discussed the possibility for zonule tension influencing the subcellular distribution and activity of Piezo1 channel similar to the TRPV4 channels in the mouse lens.

Response to Reviewer 2:

Thank you for submitting your response. The following questions needs to be resolved which will greatly improve the scientific merit of this paper.

Comment 1: The tension loading experiments need to be carried out for 1hr and then you will need to test for the phosphorylation state of MLC. Once the result is obtained then you will be in position to infer correlation between tensile forces and activation of MLC and further interrelation with Piezo 1 activity.

Response: We thank the reviewer’s suggestion, however, our goal in this study was to determine whether lens shape change induced by force or load influences the biochemical changes related to the contraction and relaxation. For this, since the lens shape change in vivo is dynamic and rapid, we tested 60 seconds load response. In this study, we did not focus our effort to correlate the load-induced changes with Piezo1 channel induced changes. Instead, we investigated these aspects separately. In future studies, it is important to investigate whether load influences the Piezo1 channel activity and understanding the consequences of Piezo activation on lens shape change.

Comment 2: You will need to acquire and test an alternative antibody/antibodies that work on the lens to determine the expression of Piezo 2 channel.

Response: We are not ruling out the expression of Piezo2 in lens but our findings from the RNAseq-based expression profile and RT-PCR and qRT-PCR amplification reveal that the relative expression level of Piezo2 is much less than Piezo1 in mouse lens. Therefore, we decided not to use any additional resources since we also did not find any convincing results in the immunoblot blot analysis performed using one of the Piezo-2 antibodies procured from the commercial sources (Fig.S2). Additionally, we have used the same antibody with human trabecular meshwork cell lysates and detected the Piezo-2 specific protein bands and this date were published (Maddala & Rao, Am J Physiol Cell Physio, l319: C288–C299, 2020).  

Comment 3:You will need to characterize the subcellular expression of Piezo 1 in wild type P30 lenses by showing the immnulableing result from of the outer cortex, inner cortex and core regions of the lens using high magnification images in equatorial orientation and present the same information when using the PiezotdT model (P30 lens, equatorial section, outer cortex, inner cortex and core) so that an arbitrary comparison can be made in the expression pattern of Piezo 1 and if it matches between the two mouse types.

Response: We thank the reviewer for this recommendation to perform additional analyses. However, unfortunately, right now we are not in a position to perform these recommended additional analyses to have a comprehensive subcellular distribution profile of Piezo channels in lens.

Comment 4:You will need to perform an antibody specifity reaction using an antigenic control reaction to determine which of the bands in your western are specific.

Response: As we explained previously, since piezo1 antibody produced multiple bands using the lysates derived from the wild type mouse lens, we procured the piezo1-tdT transgenic mice to confirm the distribution and expression of the fusion protein of Piezo1 and tdT in the lens tissue. The transgenic lens lysates also produced multiple bands indicating that there is a propensity for proteolytic cleavage of Piezo1 channel in the lens fibers. We discussed this aspect in the Discussion section as also recommended by Reviewer 1. Additionally, we have reviewed the published data and the datasheet information of Piezo1 antibodies from various commercial sources and the antibodies of Piezo1 react with multiple protein bands including a prominent band at round 70-75 kDa. Therefore, we are confident that our results and conclusions are consistent.  

Comment 5: Your P1 western (Fig 3C) is showing a very week expression of Piezo 1 at the ~250kda band while the intensity increases in P14 and P16 but decreasing in P21 and P27 with an increased amount of degradation product at ~75kDa bands. This result suggests that either the level of expression of Peizo 1 changes throughout postnatal development or that the samples are not handled well and the protein degrades. The characterization of expression during postanatal development using western blotting analysis is interesting but is lacking any connection with the topic of this paper. Your western blot of P1 shows a week band of expression of Peizo 1 but your localization expression using immonolabeling of P1 lenses (Fig 3 E) shows a good signal. Therefore, you will need to work on this discrepancy and show additional image of the expression of P1 using western blot.

Response: We understand clearly the reviewer’s concern and comment regarding the posttranslational modification (proteolytic cleavage) of piezo1 in the lens tissue and its distribution pattern in the differentiating and differentiated fibers. Although, our data requires additional rigor to understand the definitive posttranslational changes of Piezo1 channel in lens, this exploratory data reveal this possibility, and requirement for additional studies in future. The immunofluorescence data shown to Fig. 3 is qualitative and cannot be compared directly with Western blot data. However, we have revised Fig. 3 to show the background staining detected with a secondary antibody alone (Fig. 3G), and compared to this control, the staining detected with Piezo1 antibody is much more intense and specific.   

Comment 6. You will need to identify the pattern of expression/localization in subcellular level in postnatal stages which will match the stages in your western blotting although this is not in the scope of the topic of this paper.

Response: We recognize the importance of recommended aspects regarding the Piezo1 channel’s expression, subcellular distribution and stability during lens differentiation and maturation. These aspects have to be explored in future studies, and unfortunately, currently, we do not have resources and personnel to perform the recommended comprehensive analyses.

Reviewer 2 Report

Thank you for submitting your response. The following questions needs to be resolved which will greatly improve the scientific merit of this paper.

  1. The tension loading experiments need to be carried out for 1hr and then you will need to test for the phosphorylation state of MLC. Once the result is obtained then you will be in position to infer correlation between tensile forces and activation of MLC and further interrelation with Piezo 1 activity.

  1. You will need to acquire and test an alternative antibody/antibodies that work on the lens to determine the expression of Piezo 2 channel.

  1. You will need to characterize the subcellular expression of Piezo 1 in wild type P30 lenses by showing the immnulableing result from of the outer cortex, inner cortex and core regions of the lens using high magnification images in equatorial orientation and present the same information when using the PiezotdT model (P30 lens, equatorial section, outer cortex, inner cortex and core) so that an arbitrary comparison can be made in the expression pattern of Piezo 1 and if it matches between the two mouse types.

  1. You will need to perform an antibody specifity reaction using an antigenic control reaction to determine which of the bands in your western are specific.

  1. Your P1 western (Fig 3C) is showing a very week expression of Piezo 1 at the ~250kda band while the intensity increases in P14 and P16 but decreasing in P21 and P27 with an increased amount of degradation product at ~75kDa bands. This result suggests that either the level of expression of Peizo 1 changes throughout postnatal development or that the samples are not handled well and the protein degrades. The characterization of expression during postanatal development using western blotting analysis is interesting but is lacking any connection with the topic of this paper. Your western blot of P1 shows a week band of expression of Peizo 1 but your localization expression using immonolabeling of P1 lenses (Fig 3 E) shows a good signal. Therefore, you will need to work on this discrepancy and show additional image of the expression of P1 using western blot.

  1. You will need to identify the pattern of expression/localization in subcellular level in postnatal stages which will match the stages in your western blotting although this is not in the scope of the topic of this paper.

Author Response

(The authors gave the same response as above.)

Round 3

Reviewer 1 Report

The authors have addressed the majority of my concerns and have added supplementary data. They have also expand the limitations of their study due to the effects of  the COVID pandemic and the availability of animal lines. However, I still do not believe that these circumstances should be a reason for presenting in complete data around the localization of Piezo 1 in the. This incomplete data set tends to diminish the important finding that links Piezo 1 activation as a transducer of changes in the tension applied to the lens and the MLC kinase activation to presumable alter the lens cytoskeleton that determines lens shape.

Hence it is still my recommendation to remove the immunolabelling from the lenses expressing the Piezo1-tdT fusion protein (Figure 4C) as this data set is incomplete and does not map the distribution of Piezo 1 through out all regions of the lens. If the authors still insist on retaining Figures 3E-G they should indicate that this is a preliminary mapping of Piezo 1 in the mouse lens and that a more detailed characterization of regional difference in the subcellular expression of Piezo 1 labelling is required. 

As a tool to verify Piezo 1 identification in the lens the authors could still use the Western blot  Piezo1-tdT fusion (Figure 4A &B) but the blot should use the same size range as used in Figure 3C & D to facilitate comparison and to evaluate whether a 75kDa band is also seen in the blot for Piezo1-tdT fusion